# A meiosis-specific BRCA2 binding protein recruits recombinases to DNA double-strand breaks to ensure homologous recombination

Jingjing Zhang[1], Yasuhiro Fujiwara[2], Shohei Yamamoto[3] & Hiroki Shibuya[1]

Homologous recombination (HR) repairs DNA double-strand breaks (DSBs) to maintain genomic integrity. Recombinase recruited to the DSBs by the mediator protein BRCA2 catalyzes the homology-directed repair. During meiotic HR, programmed DSBs are introduced genome-wide but their repair mechanisms, including the regulation of BRCA2, have remained largely elusive. Here we identify a meiotic localizer of BRCA2, MEILB2/HSF2BP, that localizes to the site of meiotic DSBs in mice. Disruption of *Meilb2* abolishes the localization of RAD51 and DMC1 recombinases in spermatocytes, leading to errors in DSB repair and male sterility. MEILB2 directly binds to BRCA2 and regulates its association to meiotic DSBs. We map the MEILB2-binding domain within BRCA2 that is distinct from the canonical DNA-binding domain but is sufficient to localize to meiotic DSBs in a MEILB2-dependent manner. We conclude that localization of BRCA2 to meiotic DSBs is mediated by MEILB2, which is an integral mechanism to repair abundant meiotic DSBs.

[1] Department of Chemistry and Molecular Biology, University of Gothenburg, SE-40530 Gothenburg, Sweden. [2] Institute for Quantitative Biosciences, University of Tokyo, 1-1-1 Yayoi, Tokyo 113-0032, Japan. [3] Graduate Program in Bioscience, Graduate School of Science, University of Tokyo, Hongo, Tokyo 113-0033, Japan. These authors contributed equally: Jingjing Zhang, Yasuhiro Fujiwara. Correspondence and requests for materials should be addressed to H.S. (email: hiroki.shibuya@gu.se)

DNA lesions threaten genomic integrity by interfering with a wide range of cellular processes, such as DNA replication, DNA transcription, and chromosome segregation[1]. Improperly repaired DNA lesions ultimately lead to genomic rearrangements, a hallmark of cancer cells[2]. DNA double-strand breaks (DSBs) are the most cytotoxic DNA lesions, and these are repaired mainly by two alternative pathways, the non-homologous end joining and the homologous recombination (HR) pathways[3,4]. HR uses the intact sister chromatid as a repair template and therefore is the more error-free pathway, and this is especially important for the maintenance of genomic integrity and the prevention of tumor development[5].

An important HR gene is breast cancer susceptibility gene 2 (*BRCA2*), and germline mutation of this gene is a major risk factor for the development of human breast and ovarian cancers[6,7]. After DSB formation, the break sites are resected into single-strand DNA (ssDNA) by the MRE11-RAD50-NBS1 exonuclease complex, and the ssDNAs are coated by the highly abundant RPA ssDNA-binding complex[8]. BRCA2 subsequently removes RPA from ssDNA and recruits the RAD51 recombinase through direct binding to RAD51 and ssDNA, thus promoting RAD51 nucleoprotein filament formation on the ssDNA[9,10]. The RAD51 nucleoprotein filament subsequently catalyzes DNA strand exchange between sister chromatids in order to perform homology-directed repair. Mutation of *BRCA2* disrupts these HR processes and forces cells to repair the DSBs by more error-prone pathways, which threatens genomic integrity[6].

In addition to repairing the accidental DNA lesions in mitotic cells, HR is also important for the normal progression of meiosis[11,12]. During meiotic prophase I, HR takes place using homologous chromosomes as the primary repair template rather than sister chromatids[13], resulting in the formation of crossover structures between homologous chromosomes. Meiotic HR increases genetic diversity, promotes evolution, and, more crucially, ensures the correct segregation of homologous chromosomes during the following cell division[14]. One of the major differences between mitotic and meiotic HR resides in the DSB induction step: mitotic DSBs are introduced by accident, while meiotic DSBs are intentionally introduced by the activation of the meiosis-specific endonuclease SPO11 at the beginning of meiotic prophase I (the leptotene to zygotene stage)[15–17]. Moreover, the meiotic programmed DSBs are abundantly distributed throughout the genome (about 300 per nucleus in mice) and are all quickly repaired by the mid-pachytene stage. The repair of meiotic DSBs requires the coordinated action of two distinct recombinases, including RAD51 and its meiosis-specific paralog DMC1[18–20]. Studies in yeast suggest that the recombinase activity of DMC1 is required for the strand-exchange reaction, while RAD51 is suggested to function as an accessory factor that facilitates the localization of DMC1 onto the ssDNA[21]. DMC1 also switches the repair template from sister chromatids to homologous chromosomes, creating the so-called homolog bias that is specific to meiosis[22]. These findings suggest that DMC1, with the aid of RAD51, plays a central role in repairing meiotic DSBs.

Despite its well-established role in somatic cells as a potent cancer suppressor, the role of BRCA2 in meiotic HR is less well defined, partly due to the embryonic lethality of *Brca2* mutant animals[23]. However, in vitro studies suggest that BRCA2 directly binds to DMC1 and stimulates its recombinase activity[24,25]. Also, studies in the plant *Arabidopsis thaliana* and the worm *Caenorhabditis elegans* showed that hypomorphic mutations of *Brca2* homologs lead to errors in meiotic HR in vivo[26,27]. In the mammalian case, *Brca2* knockout (KO) mice carrying a bacterial artificial chromosome with the human *BRCA2* gene rescued the embryonic lethality but led to male sterility due to meiotic HR errors[28]. In all organisms studied, the localization of recombinases to the meiotic DSBs is impaired in the presence of *Brca2* mutations. Together these studies suggest the conserved function of BRCA2 as a recombinase recruiter in meiotic HR. However, the detailed molecular regulation of the assembly of the recombinase complexes and the role of BRCA2 in meiotic DSBs has remained poorly understood.

In this study, we have identified a germ cell-specific BRCA2-binding protein in mice, which we termed meiotic localizer of BRCA2 (MEILB2), by utilizing the previously established in vivo electroporation technique[29]. We show here that MEILB2 is a master regulator of meiotic recombinases and the localization of RAD51 and DMC1 at meiotic DSBs is completely abolished in *Meilb2* KO male mice, leading to errors in meiotic DSB repair and subsequent sterility. *Meilb2* KO female mice also show similar phenotypes but milder than males, and they have a massive reduction in the number of oocytes and suffer from subfertility. We also show that MEILB2 binds directly to BRCA2 and is responsible for BRCA2 localization at the meiotic DSBs, and this accounts for the impaired recombinase localization observed in *Meilb2* KO mice. Our findings highlight the meiosis-specific BRCA2 recruitment mechanism at the sites of DSBs, which ensures the accumulation of the meiotic recombinases needed to repair meiotic DSBs.

## Results

**MEILB2/HSF2BP is a germ cell-specific chromosomal protein.** In order to identify factors regulating meiotic DSB repair, we examined the subcellular localizations of functionally uncharacterized proteins that are upregulated in murine germ line tissues[30]. We utilized the in vivo electroporation technique[29] and expressed their green fluorescent protein (GFP)-fusion proteins in testis. One of the candidate genes of unknown function, *4932437G14Rik*, also known as *Heat shock factor 2-binding protein* (*Hsf2bp*), showed a characteristic localization pattern specifically in early prophase I spermatocytes (zygotene and early pachytene stages) in which punctate signals were formed along the chromosome axes similar to the distribution of meiotic recombination nodules (Fig. 1a). This protein was previously identified as a binding protein of heat shock factor 2 (HSF2) by the yeast two-hybrid (Y2H) screening of HSF2; however, its physiological function has not been addressed[31]. According to the molecular function described below, we renamed this protein as MEILB2. MEILB2 is a 338 amino acid (a.a.) protein composed of an N-terminus coiled-coil domain and a C-terminus armadillo repeat domain (composed of four armadillo repeats) (Fig. 1b). BLAST database searches identified MEILB2 homologs that are widely conserved in vertebrate species (Supplementary Fig. 1). By reverse transcription PCR (RT-PCR) in mice, we confirmed that *Meilb2* mRNA was upregulated in germline tissues such as testis and embryonic ovary and was barely expressed in the other somatic tissues (Fig. 1c). Taken together, we redefined MEILB2 as a meiotic prophase I-specific chromosomal axis-associating protein.

**MEILB2 localizes to meiotic recombination sites.** To determine the subcellular localization of endogenous MEILB2, we generated polyclonal antibodies against MEILB2 and used them for immunostaining of spermatocyte chromosomal spreads. Consistent with the localization of GFP-fusion protein, endogenous MEILB2 showed punctate localization along chromosome axes specifically in early meiotic prophase I cells (Fig. 1d). MEILB2 foci started to appear from the leptotene stage, reached their greatest number in the zygotene stage (257 foci on average), persisted until the early pachytene stage, and finally disappeared

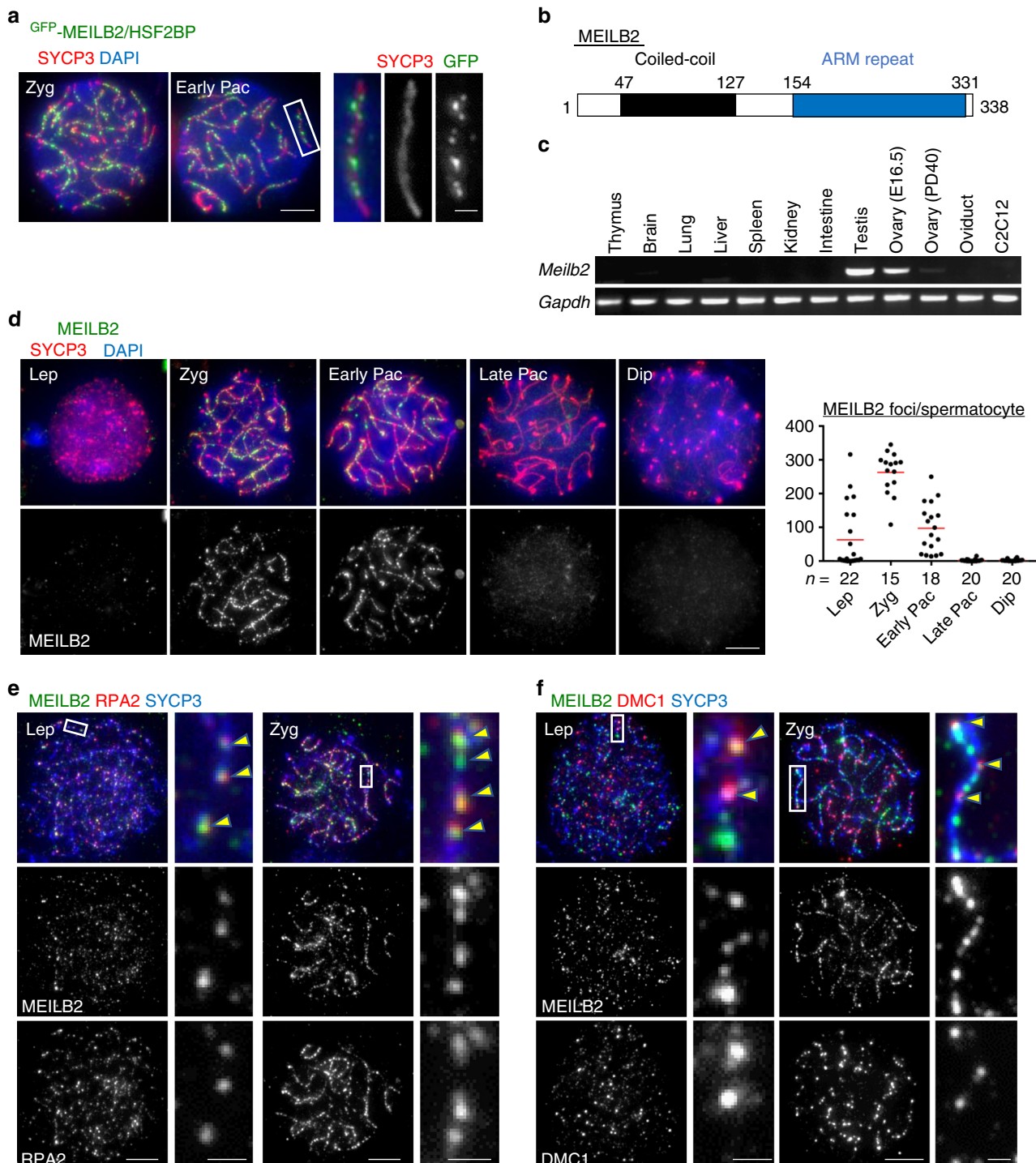

**Fig. 1** Identification of MEILB2 as a meiotic chromosomal protein. **a** Wild-type (WT) spermatocytes expressing GFP-MEILB2/HSF2BP stained with the indicated antibodies and 4,6-diamidino-2-phenylindole (DAPI). **b** The domain conformation of MEILB2. The coiled-coil domain (a.a. 47–127) and armadillo (ARM) repeat domain (a.a. 154–331) are shown. **c** Tissue-specific expression levels of *Meilb2* and *Gapdh* (a loading control). C2C12 is a mitotically rounding cancer cell line. **d** WT spermatocytes stained with the indicated antibodies and DAPI. Each meiotic prophase I substage is shown. The graph shows the number of MEILB2 foci associated with the chromosome axes. The mean value is shown as a red bar. *n* shows the analyzed spermatocyte number pooled from two mice. *Lep* leptotene (dotty or discontinuous SYCP3), *Zyg* zygotene (linear SYCP3 with partial synapsis), *Pac* pachytene (Early; linear SYCP3 with complete synapsis and Late; linear SYCP3 with complete synapsis and thickened SYCP3 ends), *Dip* diplotene (linear SYCP3 with desynapsis). **e, f** WT spermatocytes stained with the indicated antibodies and DAPI. The co-localizing foci along the chromosome axis are highlighted by yellow arrowheads. The quantification of co-localization was performed using three late leptotene cells and six zygotene cells pooled from two mice for RPA2 (**e**) and six late leptotene cells and ten zygotene cells pooled from three mice for DMC1 (**f**). The axis-associated foci are counted. Scale bars, 5 and 1 μm (magnified panel). Source data are provided as a Source Data file

in the late pachytene stage. The majority of the MEILB2 foci co-localized with the ssDNA-binding protein RPA2, which marks the site of resected DNA at the DSB sites, suggesting that MEILB2 foci largely corresponded to DSB sites (Fig. 1e; 75% and 84% of the RPA2 foci stained positive for MEILB2 and 85% and 85% of the MEILB2 foci stained positive for RPA2 at the late leptotene and zygotene stages, respectively). We next stained for the recombinase DMC1, which is another DSB marker. It is reported that the localization of DMC1 onto DSB sites is more temporally restricted compared to that of RPA2, most likely due to the quick removal of DMC1 after the completion of the strand invasion event[32]. Consistent with this notion, MEILB2 foci were more abundant than DMC1 foci, and significantly, most of the DMC1 foci accompanied MEILB2 foci (Fig. 1f; 65% and 81% of the DMC1 foci stained positive for MEILB2 and 17% and 31% of the MEILB2 foci stained positive for DMC1 at the late leptotene and zygotene stages, respectively). Further, in the early pachytene stage, the DMC1 foci mostly disappeared from autosomes and became restricted to the sex chromosomes (Supplementary Fig. 2a), while MEILB2 foci were still abundant even along autosomes, supporting the notion that MEILB2 remained on the recombination nodules even after the removal of DMC1 (Supplementary Fig. 2a). We also confirmed the partial co-localization of MEILB2 and RAD51 (Supplementary Fig. 2b; 68% and 76% of the RAD51 foci stained positive for MEILB2 and 29% and 42% of the MEILB2 foci stained positive for RAD51 at the late leptotene and zygotene stages, respectively), which was consistent with the notion that DMC1 and RAD51 co-localize with each other[33]. Together, these data suggest that MEILB2 is a recombination nodule protein that associates with the DSB sites similarly to RPA2 and in a manner that is more stable and persistent than DMC1 and RAD51.

**MEILB2 binds to BRCA2**. To identify proteins that directly bind to MEILB2, we conducted a comprehensive Y2H screening in a mouse testis cDNA library. Intriguingly, BRCA2 was repeatedly identified as a MEILB2-binding protein (Fig. 2a). Three unique peptides of BRCA2, located between the N-terminus BRC repeats and the C-terminus ssDNA-binding domain, were identified (Fig. 2a). We termed the short 223 a.a. sequence of BRCA2, which was common in all three unique peptides, as a MEILB2-binding domain (MBD) (Fig. 2a and Supplementary Fig. 3). We confirmed that both the MBD alone and the C-terminus (containing the MBD and the ssDNA-binding domain) of BRCA2 indeed bind to MEILB2 in the Y2H system (Fig. 2b).

We also verified this interaction using a mouse cell line co-expressing GFP-BRCA2 truncations (GFP-BRCA2-N, M, and C) with MEILB2-MYC. The GFP pull-down assay detected the specific interaction between GFP-BRCA2-C and MEILB2-MYC (Fig. 2c). Further, GFP-BRCA2-C without the MBD (GFP-BRCA2-C ΔMBD) abolished this interaction, proving that the MBD of BRCA2 was necessary and sufficient for the MEILB2 interaction (Fig. 2d). Notably, endogenous RAD51 was pulled-down with GFP-BRCA2-M containing BRC repeats, which is consistent with the previous study (Fig. 2c)[9], and which proves that MEILB2 and RAD51 bind to distinct BRCA2 domains.

To verify the interaction in vivo, we performed MEILB2 immunoprecipitation from mouse testis extracts and blotted the immunoprecipitates with a BRCA2 antibody that had been generated in a previous study[34]. Consistent with the Y2H and pull-down interactions, we detected quite efficient co-immunoprecipitation of BRCA2, concluding that BRCA2 is a physiological binding partner of MEILB2 (Fig. 2e). Notably, we also detected the co-immunoprecipitation of RAD51 in MEILB2

immunoprecipitates (Fig. 2e), likely through an indirect interaction mediated by BRCA2. In contrast, DMC1 was not co-immunoprecipitated with MEILB2, which might reflect the reported weaker DMC1–BRCA2 interaction compared to RAD51–BRCA2 (Fig. 2e)[24,25].

**Meilb2−/− male shows prophase I arrest with synapsis defects**. To address the function of MEILB2, we made Meilb2 KO mice using a gene-targeted embryonic stem cell line (Fig. 3a). The western blot using testis extracts indicated that MEILB2 protein expression was indeed abolished in our homozygous (Meilb2−/−) mice (Fig. 3b). The Meilb2−/− mice showed normal development with no overt somatic phenotype but exhibited complete infertility in males, and male adult mice had smaller testes compared to their wild-type (WT) littermates (Fig. 3c). The juvenile testes at postnatal day (PD) 14 showed no size difference between Meilb2−/− and WT, suggesting that the defects likely occur after meiotic entry (Supplementary Fig. 4a). Further histological analysis confirmed the presence of spermatogonia and meiotic prophase I cells at the periphery of the seminiferous tubules but the absence of spermatids at the center of the seminiferous tubules in Meilb2−/− testes (Fig. 3d). The TdT-mediated dUTP nick-end labeling (TUNEL) assay showed large numbers of apoptotic cells at the periphery of the Meilb2−/− seminiferous tubules (Fig. 3e), suggesting that the germ cells died during the progression of meiotic prophase I.

To determine the cellular defects accounting for the sterility observed in male mice, we studied chromosome spreads of Meilb2−/− spermatocytes. First, we confirmed that punctate foci at DSB sites, stained by our MEILB2 antibody, were totally absent in Meilb2−/− spermatocytes (Fig. 3f), which was consistent with the undetectable protein expression in western blots (Fig. 3b). To measure meiotic prophase I progression, we stained the spermatocytes with the chromosomal synapsis marker SYCE3, which marks the synapsed chromosome axes[35]. While WT spermatocytes achieved complete homologous synapsis and reached the pachytene stage, Meilb2−/− spermatocytes were arrested at the zygotene stage, with incomplete synapsis, and they never reached the pachytene or later stages (Fig. 3g). We further confirmed the cell cycle arrest at early prophase I stage in Meilb2−/− spermatocytes by the absence of histone H1T staining (Fig. 3h), which starts to appear from the mid-pachytene stage in WT spermatocytes[36]. We noticed that the zygotene-arrested Meilb2−/− spermatocytes frequently exhibited the partner switch phenotype (Fig. 3i), which is indicative of the aberrant nonhomologous synapsis that has been reported in several recombination mutant mice[37,38]. These data suggest that in the absence of MEILB2 spermatocytes are arrested in the zygotene stage with incomplete synapsis and aberrant nonhomologous synapsis, which leads to male sterility.

**MEILB2 is dispensable for the introduction of meioitc DSBs**. Proper homologous synapsis is ensured by the DNA homology search between homologous chromosomes during the strand invasion step of meiotic HR[11,12]. To determine whether or not DSBs are introduced normally in Meilb2−/− spermatocytes, we stained the nuclei with a DNA damage marker, phosphorylated histone H2AX (γH2AX)[39]. In WT spermatocytes, the γH2AX signal became detectable in the leptotene stage with the induction of programmed DSBs, increased in the zygotene stage, and then became restricted to the sex chromosomes in the pachytene and diplotene stages along with the gradual repair of the DSBs (Fig. 4a). Also, in Meilb2−/− spermatocytes the γH2AX signal appeared normally in the leptotene stage and persisted until the spermatocytes were arrested at the zygotene stage (Fig. 4b and

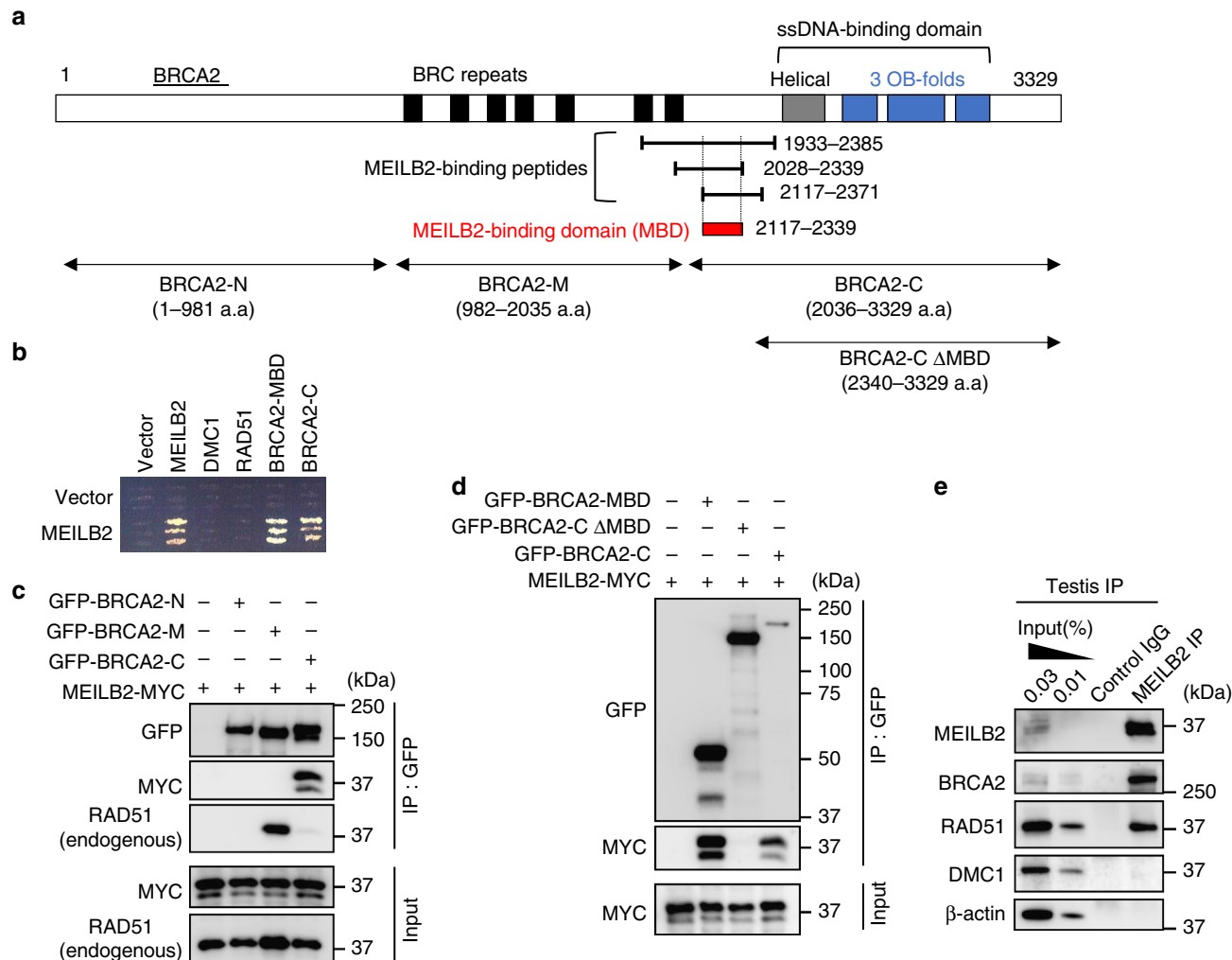

**Fig. 2** Identification of BRCA2 as a MEILB2-binding protein. **a** The domain conformation of BRCA2. The BRC repeat domains, helical domain, and OB-fold domains are shown. Three unique peptides (a.a. 1933–2385, 2028–2339, and 2117–2371) identified in the yeast two-hybrid screening as MEILB2-binding peptides are shown. The common domain (a.a. 2117–2339) in all three peptides is also shown as the MEILB2-binding domain (MBD). **b** Yeast two-hybrid interactions between MEILB2 (prey) and MEILB2, DMC1, RAD51, BRCA2-MBD, and BRCA2-C (bait). **c, d** Immunoprecipitates with the green fluorescent protein (GFP) antibody from B16-F1 cells expressing MEILB2-MYC and GFP-BRCA2 truncations N (a.a. 1–981), M (a.a. 982–2035), or C (a.a. 2036–3329) in **c** and MBD (a.a. 2117–2339), C ΔMBD (a.a. 2340–3329), or C (a.a. 2036–3329) in **d**. Input and immunoprecipitates (IP) were immunoblotted with the indicated antibodies. **e** IP from mouse testis extracts with the MEILB2 antibody or with IgG as the negative control and immunoblotted with the indicated antibodies. The blots with β-actin served as the loading control

Supplementary Fig. 4c). Consistent with this, RPA2 staining appeared normally and accumulated toward the zygotene stage in $Meilb2^{-/-}$ spermatocytes in a manner comparable to WT spermatocytes (Fig. 4c and Supplementary Fig. 4d). Together these results suggest that DSBs are introduced normally and are resected into ssDNA but are not repaired in $Meilb2^{-/-}$ spermatocytes.

**MEILB2 is indispensable for the recruitment of recombinases**. After the introduction of DSBs and their resection into ssDNA, the recombinases RAD51 and DMC1 are recruited to the sites of DSBs, and these form nucleoprotein filaments on ssDNA and promote strand invasion[17,40]. In WT spermatocytes, we observed that both DMC1 and RAD51 foci started to appear in the leptotene stage, reached their maximum number in the zygotene stage, and gradually disappeared toward the late pachytene stage (Fig. 4d, e and Supplementary Fig. 5a), which was consistent with the results of previous studies[32]. To our

surprise, however, the localization of DMC1 and RAD51 in $Meilb2^{-/-}$ spermatocytes was almost totally abolished throughout the leptotene to zygotene stages (Fig. 4d, e and Supplementary Fig. 5a). We confirmed that the protein expression of DMC1 and RAD51 was comparable between WT and $Meilb2^{-/-}$, proving that MEILB2 is needed for the localization, but not for the expression, of recombinases (Fig. 4f). The staining of MLH1, a marker of sites that are destined to become crossovers, confirmed the total abolishment of crossover formation in $Meilb2^{-/-}$ spermatocytes (Supplementary Fig. 5b), consistent with the cell cycle arrest at the zygotene stage due to the mislocalization of recombinases.

The staining of SPATA22, which associates with the recombination intermediates by forming a complex with the meiosis-specific ssDNA-binding protein MEIOB[41–43], showed that the loading of SPATA22 occurred normally in the leptotene stage but the foci abnormally accumulated toward the zygotene stages in $Meilb2^{-/-}$ spermatocytes compared to the WT (Fig. 4g). Further, the signal intensity of SPATA22 in $Meilb2^{-/-}$ zygotene

spermatocytes was significantly higher than WT (Supplementary Fig. 5c). These data suggest that the incompletely repaired recombination intermediates, which are associated with SPATA22, accumulated in *Meilb2*[−/−] spermatocytes. We also detected the in vivo interaction between MEILB2 and SPATA22, but not the RPA complex, by MEILB2 immunoprecipitation implying some potential functional interplay between MEILB2 and SPATA22 (Supplementary Fig. 5d).

To clarify the localization hierarchy, we stained MEILB2 in *Spo11*[−/−] and *Dmc1*[−/−] spermatocytes and found that the punctate localization of MEILB2 along the chromosome axes was totally abolished in *Spo11*[−/−] (Fig. 4h, middle) but not in *Dmc1*[−/−] spermatocytes (Fig. 4h, right). These results suggest that MEILB2 functions after the induction of DSBs but before the recruitment of DMC1 and the following strand invasion event. Collectively, we conclude that MEILB2 is required for the

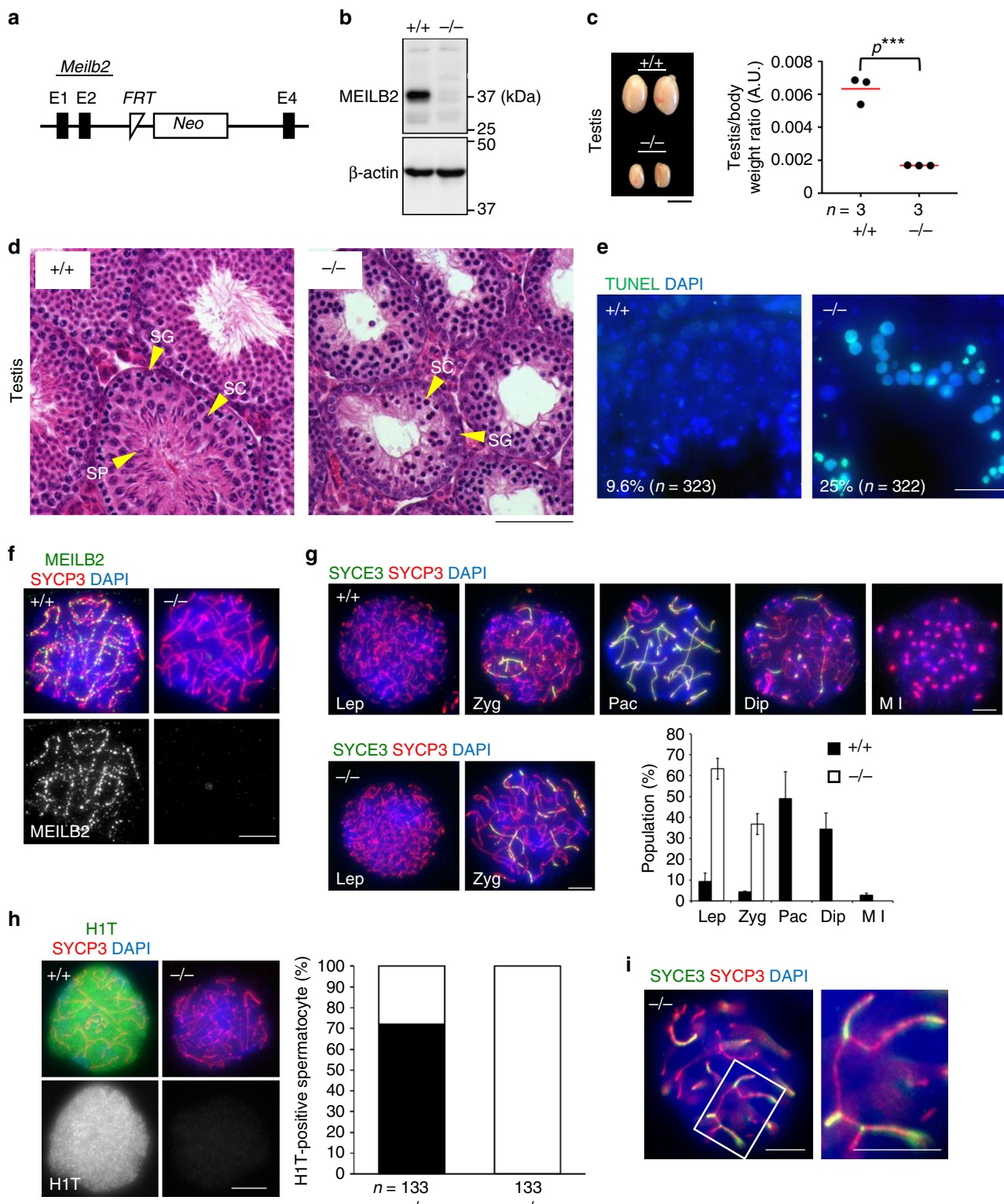

**Fig. 3** Infertility in *Meilb2*[−/−] male mice. **a** Schematic of the *Meilb2* KO allele. *E* exon, *FRT* flippase recognition site, *Neo* neomycin cassette. **b** Immunoblots of testis extracts from wild-type (WT) (+/+) and *Meilb2* KO (−/−) with the indicated antibodies. **c** Testes from WT (+/+) and *Meilb2* KO (−/−) at 4 months of age with the quantification of testis/body weight ratio. The mean value is shown as a red bar. *n* shows the analyzed mouse number. Scale bar, 2 μm. **d** Testis sections from 8-week-old WT (+/+) and *Meilb2* KO (−/−) stained with hematoxylin and eosin. *SG* spermatogonia, *SC* spermatocyte, *SP* spermatid. Scale bar, 100 μm. **e** Testis sections from 8-week-old WT (+/+) and *Meilb2* KO (−/−) stained with TdT-mediated dUTP-fluorescein nick end labeling (TUNEL) and 4,6-diamidino-2-phenylindole (DAPI). TUNEL-positive seminiferous tubules (those containing more than three TUNEL-positive cells) were quantified. *n* shows the analyzed seminiferous tubule number pooled from two mice. Scale bar, 15 μm. **f** Zygotene spermatocytes from WT (+/+) and *Meilb2* KO (−/−) stained with the indicated antibodies and DAPI. Scale bar, 5 μm. **g** Spermatocytes from WT (+/+) and *Meilb2* KO (−/−) stained with the indicated antibodies and DAPI. SYCP3-positive spermatocytes (1083 cells for WT and 1128 cells for KO, pooled from two mice for each genotype) were classified into the following substages: Lep (leptotene; no SYCE3); Zyg (zygotene; partially assembled SYCE3); Pac (pachytene; fully assembled SYCE3); Dip (diplotene; disassembled SYCE3); and Met I (metaphase I; SYCP3 accumulations at centromeres). The mean values of two independent experiments from two different mice are shown. Error bars show the SD. Scale bars, 5 μm. **h** Pachytene spermatocytes from WT (+/+) and zygotene-arrested spermatocytes from *Meilb2* KO (−/−) stained with the indicated antibodies and DAPI with the quantification of H1T-positive spermatocyte ratio. *n* shows the analyzed spermatocyte number pooled from two mice for each genotype. Scale bar, 5 μm. **i** Zygotene spermatocytes from *Meilb2* KO (−/−) stained with the indicated antibodies and DAPI. About 92% of *Meilb2* KO zygotene spermatocytes (25 zygotene cells pooled from three mice) showed at least one partner switch (magnified). Scale bars, 5 μm. All analyses were with two-tailed *t* tests. ***$p < 0.001$. Source data are provided as a Source Data file

recruitment of recombinases onto ssDNA after the induction of DSBs.

**The *Meilb2* knockout phenotype is sexually dimorphic.** Our RT-PCR experiment showed that *Meilb2* mRNA is also upregulated in the embryonic ovary undergoing meiotic prophase I, suggesting that MEILB2 also functions in female meiosis (Fig. 1c). The staining of prophase I oocytes with MEILB2 antibody recapitulated the punctate localization pattern specific to early prophase I oocytes similar to what was seen in spermatocytes (Fig. 5a). We confirmed that this staining completely disappeared in *Meilb2*[−/−] oocytes (Fig. 5b), leading to the conclusion that the spatiotemporal localization of MEILB2 is conserved in both sexes.

We next stained oocytes from embryonic day 19.5 (E19.5) mice with antibodies against SYCE3 to measure the progression of meiotic prophase I. In WT E19.5 mice, most of the oocytes (90%) progressed to the pachytene and diplotene stages in a semisynchronous manner, and only 6% of the oocytes were in the zygotene stage (Supplementary Fig. 6). In the *Meilb2*[−/−] ovary, however, a significant number of oocytes (24%) remained in the zygotene stage, suggesting that the progression of homologous synapsis is delayed in this mutant (Supplementary Fig. 6).

To measure the kinetics of DSB formation and repair processes in *Meilb2*[−/−] oocytes, we stained E14.5 oocytes with antibodies against RPA2, DMC1, and RAD51. RPA2 foci showed almost the same spatiotemporal distribution in *Meilb2*[−/−] oocytes as in WT controls, suggesting that the induction of DSBs and their resection occur normally in *Meilb2*[−/−] oocytes (Fig. 5c and Supplementary Fig. 7a). However, there were significant reductions in the numbers of DMC1 and RAD51 foci in *Meilb2*[−/−] oocytes compared to WT oocytes (56% and 46% reductions, respectively, in the zygotene stage), suggesting that recombinase recruitment is significantly impaired in female meiosis as well (Fig. 5d, e, and Supplementary Fig. 7a). Notably, compared to the complete disruption of RAD51 and DMC1 localization in male meiosis, the defect in females was rather mild, similar to the sexually dimorphic phenotypes reported in other meiotic recombination mutant mice[44].

In order to determine the terminal phenotype in females in the absence of *Meilb2*, we observed mature ovaries at PD25 and counted the number of follicles compared to the WT littermates. We classified the follicles into primordial, primary, and growing follicles based on their developmental stages[45]. We found a significant reduction in all three types of follicles in *Meilb2*[−/−] ovaries (75%, 54%, and 43% reductions in primordial, primary, and growing follicles, respectively), suggesting that a large portion of oocytes were eliminated during early development prior to the

formation of primordial follicles (Fig. 5f), which was likely due to the above-mentioned prophase I defects. Consistent with these results, the fertility assay showed a 40% reduction in litter size in *Meilb2*[−/−] females crossed with WT males compared to WT females (Fig. 5g). Notably, the metaphase I oocytes from *Meilb2*[−/−] adult females had the normal number of 20 bivalent chromosomes, suggesting that defective oocytes were already eliminated during prophase I during embryonic development and that the surviving oocytes found in the adult females were those that had achieved complete meiotic HR in the absence of MEILB2 (Supplementary Fig. 7b).

Based on our collective results, we concluded that MEILB2 is also required for recombinase recruitment in female meiosis, although the degree of defects in female KO mice was milder than in males, and consequently *Meilb2*[−/−] females exhibited reduced follicle formation and a subfertile phenotype.

**MEILB2 recruits BRCA2 to the meiotic recombination sites.** *Brca2* KO mice are embryonic lethal, and thus there has been no direct assessment of this gene's function in meiosis[46]. However, *Brca2*-null mice expressing human *BRCA2* rescued the embryonic lethality and showed sterility with reduced localization of recombinases similar to our *Meilb2*[−/−] mice[28]. Together with the observed MEILB2–BRCA2 interaction (Fig. 2), we hypothesized that MEILB2 regulates BRCA2 localization at meiotic DSBs. However, the meiotic localization of BRCA2 is still controversial. One human study reported that BRCA2 forms recombination nodule-like foci along chromosome axes in human spermatocytes[47], while a mouse study detected cloudy nuclear signals and failed to detect any punctate localization in murine spermatocytes[28]. Our immunostaining of murine spermatocytes using our BRCA2 polyclonal antibodies also showed a cloudy nuclear signal that was hardly distinguishable from background signal (Supplementary Fig. 8a).

We reasoned that the limited protein level of endogenous BRCA2 or limited sensitivity of the BRCA2 antibody might hinder BRCA2 detection by immunostaining. To improve the sensitivity of the detection, we overexpressed GFP-fusion constructs of *Brca2* by in vivo electroporation. While we did not observe any specific localization of GFP-BRCA2-N or GFP-BRCA2-M (Supplementary Fig. 8b), we were able to detect the recombination nodule-like foci of GFP-BRCA2-C on the chromosome axes (Fig. 6a). We also expressed the shorter MBD fragment (GFP-BRCA2-MBD), without the ssDNA-binding domain, and we observed the same spatiotemporal localization as GFP-BRCA2-C (Fig. 6b). The GFP-BRCA2-C lacking the MBD (GFP-BRCA2-C ΔMBD) showed cloudy

nuclear signals and failed to localize along chromosome axis (Supplementary Fig. 8b). Together these results suggest that the MBD is necessary and sufficient for the chromosome axis localization of BRCA2.

Importantly, these foci were specific in early prophase I cells (zygotene to early pachytene stage), which is when DSBs are

present (Fig. 6a, b). Further, the majority of GFP-BRCA2 signals co-localized with the endogenous RPA2 signals in the zygotene stage (Fig. 6c and Supplementary Fig. 8c; 73% and 68% of GFP foci were stained for RPA2 and 80% and 61% of RPA2 foci were stained for GFP in GFP-BRCA2-MBD- and GFP-BRCA2-C-expressing cells, respectively). Together these data suggest that

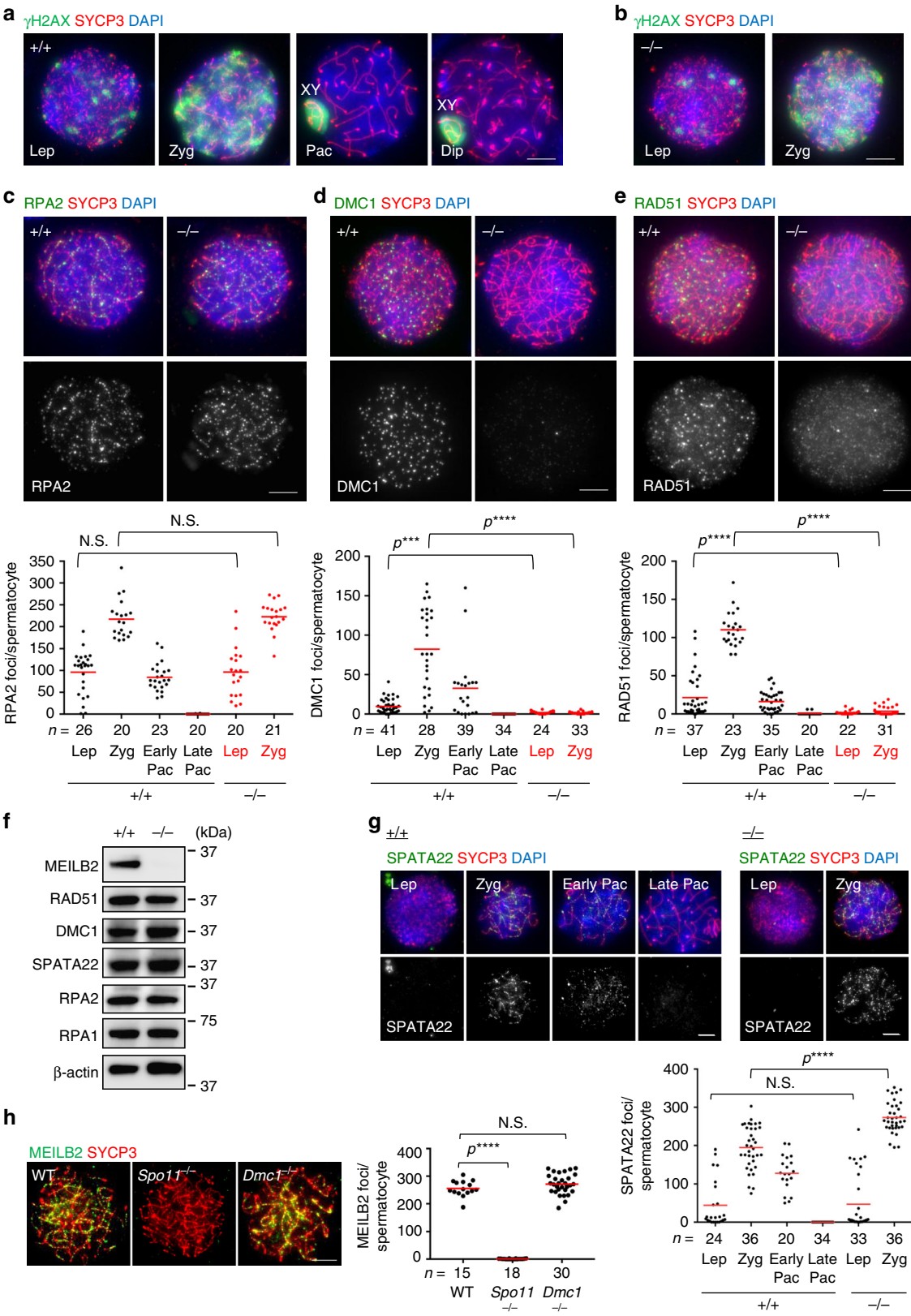

**Fig. 4** DSB repair defects in *Meilb2*−/− male mice. **a** Spermatocytes from wild-type (WT) (+/+) males stained with the indicated antibodies and 4,6-diamidino-2-phenylindole (DAPI). **b** Spermatocytes from *Meilb2* knockout (KO) (−/−) males stained with the indicated antibodies and DAPI. **c–e** Zygotene spermatocytes from WT (+/+) and *Meilb2* KO (−/−) males stained with SYCP3 in red, DAPI in blue, and RPA2 (**c**), DMC1 (**d**), or RAD51 (**e**) in green. The images in the other stages are shown in Supplementary Figs. 4d and 5a. The graph shows the number of RPA2 (**c**), DMC1 (**d**), or RAD51 (**e**) foci associated with the chromosome axes. The mean value is shown as a red bar. *n* shows the analyzed spermatocyte number pooled from three mice for each genotype. **f** Immunoblots of mouse testis extracts (PD90) from WT (+/+) and *Meilb2* KO (−/−) males using the indicated antibodies. **g** Spermatocytes from WT (+/+) and *Meilb2* KO (−/−) males stained with the indicated antibodies and DAPI. Each meiotic prophase I substage is shown. The graph shows the number of SPATA22 foci associated with the chromosome axes. The mean value is shown as a red bar. *n* shows the analyzed spermatocyte number pooled from four mice for each genotype. **h** Zygotene spermatocytes from WT, *Spo11*−/−, and *Dmc1*−/− males stained with the indicated antibodies and DAPI. The graph shows the number of MEILB2 foci associated with the chromosome axes in zygotene spermatocytes. The mean value is shown as a red bar. *n* shows the analyzed spermatocyte number pooled from two mice for each genotype. *Lep* leptotene, *Zyg* zygotene, *Pac* pachytene, *Dip* diplotene. All analyses were with two-tailed *t* tests. *N.S.* not significant. ***$p < 0.001$, ****$p < 0.0001$. Scale bars, 5 μm. Source data are provided as a Source Data file

GFP-BRCA2 localized to the DSB sites and that the MBD of BRCA2 is necessary and sufficient for this DSB localization.

To examine whether the BRCA2 localization on DSBs depends on MEILB2, we electroporated the same GFP-fusion constructs into *Meilb2*−/− testes. In line with our hypothesis, the punctate localization of both GFP-BRCA2-C and GFP-BRCA2-MBD were almost totally abolished in *Meilb2*−/− spermatocytes, and instead we observed a strong nuclear signal without any detectable foci formation (Fig. 6d, e). The abolishment of BRCA2 localization in *Meilb2*−/− spermatocytes was not because of the abnormal zygotene-stage arrest in this mutant because we could observe the punctate localization of GFP-BRCA2-C and GFP-BRCA2-MBD in the zygotene-arrested *Dmc1*−/− spermatocytes (Fig. 6d, e). Taken together, we conclude that MEILB2 functions as a recruiter of BRCA2 to the meiotic DSB sites, likely through the direct interaction between MEILB2 and BRCA2-MBD. Because BRCA2 is required for the localization of RAD51 and DMC1 in spermatocytes, the observed mislocalization of RAD51 and DMC1, as well as the consequent DSB repair defects, in *Meilb2*−/− meiocytes is likely attributable to the mislocalization of BRCA2 in this mutant (Fig. 6f).

## Discussion

A number of preceding studies have focused on the role of BRCA2 in mitotic HR as a cancer suppressor; however, meiotic regulation of BRCA2 is less well defined due to the embryonic lethality of *Brca2* KO animals[23,46]. Our study has identified a germ cell-specific binding protein of BRCA2, which we termed MEILB2, and has clarified the essential function of the MEILB2-BRCA2 complex for the successful completion of meiotic HR. Our immunoprecipitation of endogenous MEILB2 showed that MEILB2 forms complexes with RAD51 and BRCA2 in vivo. Consistent with this, the KO phenotype analyses clarified that MEILB2 functions as a recruiter of BRCA2, as well as the downstream recombinases RAD51 and DMC1, to meiotic DSBs. Notably, our *Meilb2* KO phenotypes are quite similar to the reported phenotypes in *Brca2* KO mice carrying the human *BRCA2* gene[28], confirming the coordinated function of MEILB2 and BRCA2 during meiotic DSB repair. The knockout mice of *Tex15*, a poorly characterized meiotic gene, also showed similar phenotypes in male meiosis, implying the potential functional interplay between TEX15 and MEILB2-BRCA2 complex[48]. While *Brca2* is conserved in some invertebrate species such as nematode, MEILB2 homologs are found only in vertebrate species, suggesting that MEILB2 evolved later than BRCA2 and was specialized for the meiosis-specific regulation of BRCA2.

It is known that RPA complex remains on meiotic recombination sites even after the removal of recombinases until the pachytene stage, suggesting that RPA complex somehow associates with joint molecules at this stage. Therefore, even though the

number of RPA foci is comparable between WT and *Meilb2* KO zygotene spermatocytes, these RPA foci likely represent different recombination intermediates, i.e., both unrepaired DSBs and joint molecules in WT and unrepaired DSBs in *Meilb2* KO, respectively. Further, similar to the RPA case, a significant number of MEILB2 foci remained on the recombination intermediates until early pachytene stages in WT meiocytes implying the possibility that MEILB2 also binds to joint molecules and could have some additional functions in the later stage of prophase I, such as the stabilization of joint molecules.

It is still not known why only meiotic DSB repair and not mitotic DSB repair requires MEILB2. BRCA2 has its own ssDNA-binding domain and has ssDNA-binding activity in vitro[9]. BRCA2 also binds directly to RAD51[49]. These two interactions, with ssDNA and RAD51, are sufficient for BRCA2 to recruit RAD51 onto ssDNA and to stimulate RAD51 recombinase activity in in vitro-purified systems that recapitulate the in vivo strand invasion step[9]. We argue that MEILB2 might further facilitate this pathway by recruiting BRCA2 to the ssDNA through protein–protein interactions. Indeed, our identified MBD of BRCA2, which does not contain the ssDNA-binding domain, can be targeted to the meiotic DSBs in a manner that is dependent on MEILB2. This suggests that the major pathway for recruiting BRCA2 to the meiotic DSB sites is the protein–protein interaction between the BRCA2-MBD and MEILB2 rather than the interaction between ssDNA and BRCA2. This meiosis-specific recruitment process for BRCA2, in addition to the canonical recruitment pathway through the BRCA2 ssDNA-binding domain, might ensure the efficient repair of the large number of meiotic DSBs by facilitating the localization of BRCA2 and recombinases.

It is also still not known how MEILB2 localizes to the meiotic DSB sites. We showed that MEILB2 localization to the meiotic DSBs requires SPO11, but not DMC1, suggesting that DSB formation, but not the strand invasion step, is needed for MEILB2 localization on the chromosome. We could not find any potential DNA-binding domain within the MEILB2 a.a. sequence, and it is therefore less likely that MEILB2 directly binds to ssDNA at DSBs. We speculate that there must be some scaffold protein linking MEILB2 to the ssDNA at the DSBs. It is known that ssDNA at meiotic DSBs is decorated not only with the canonical RPA complex but also with the meiosis-specific ssDNA-binding complex SPATA22-MEIOB[41–43]. Given the almost complete co-localization of MEILB2 and RPA in spermatocytes, we argue that either RPA or SPATA22-MEIOB is likely to be the scaffold for MEILB2 on ssDNA. Indeed, the localization of RPA and SPATA22 was intact in *Meilb2*−/− spermatocytes, suggesting that these ssDNA-binding complexes function epistatically to the MEILB2 localization. We also detected the in vivo interaction between MEILB2 and SPATA22, but not RPA complex, by co-immunoprecipitation. The dissection of the localization

dependency and the functional interplay between these meiotic DSB-associating proteins will be the focus of future studies designed to obtain a comprehensive picture of the DSB repair pathway in meiosis.

The sexual dimorphic phenotype has been reported in a number of meiotic recombination mutant mice, where female meiocytes always reach more advanced stages compared to those in males[44]. This is also the case for our *Meilb2* KO mice. While *Meilb2* KO male mice showed almost complete loss of RAD51

and DMC1 localizations leading to complete sterility, the female KO mice showed a reduction by almost half in the localization of these recombinases leading to subfertility (a 40% reduction in litter size). Notably, the *Meilb2* KO oocytes that managed to complete prophase I progression had a normal number of chiasmata in metaphase I, suggesting that some portion of the oocytes had achieved complete meiotic HR in the absence of MEILB2, and this is consistent with the remaining fertility in the KO female mice. The milder defects in our *Meilb2* KO female can

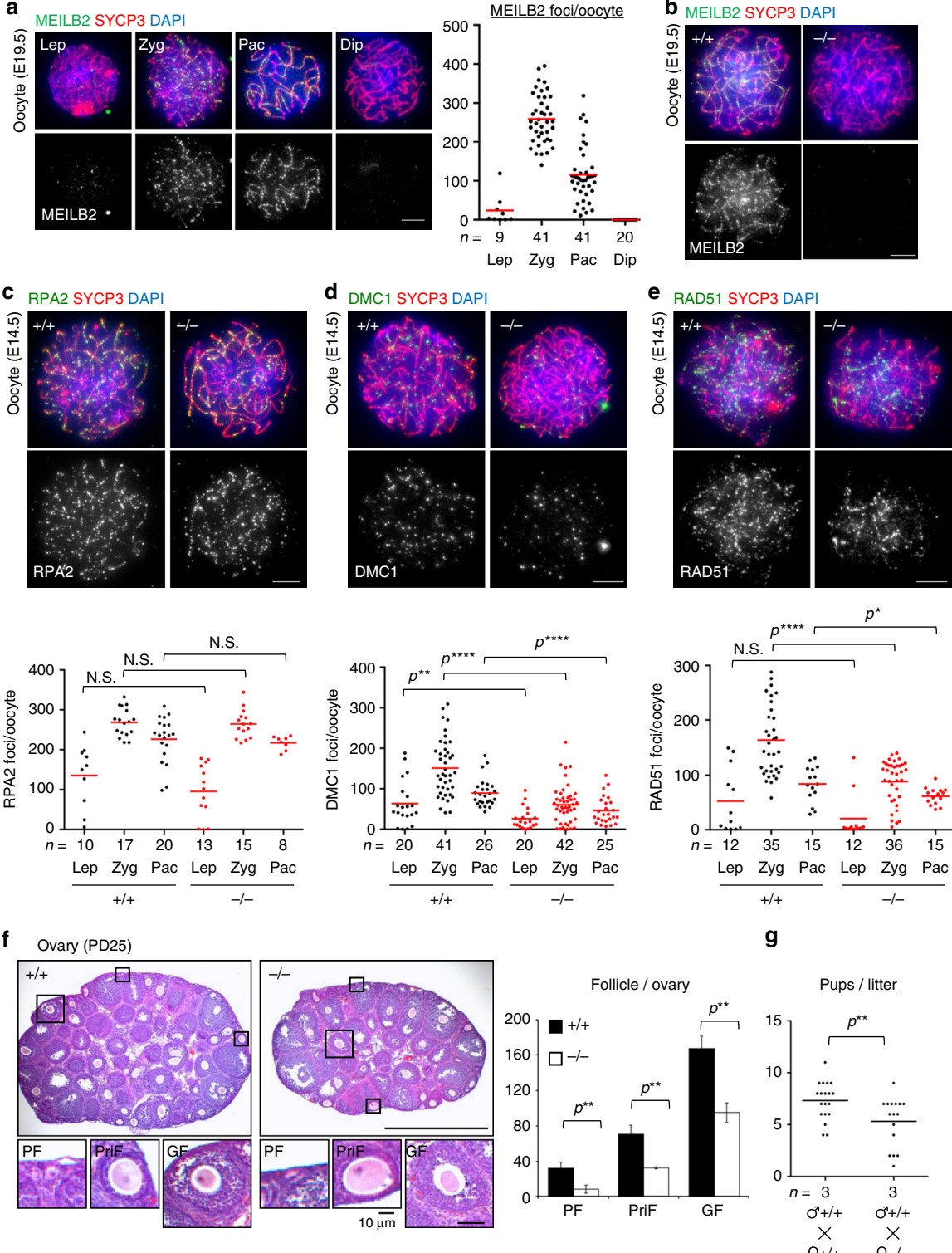

**Fig. 5** Subfertility in *Meilb2*−/− female mice. **a** Wild-type (WT) oocytes form E19.5 mice stained with the indicated antibodies and 4,6-diamidino-2-phenylindole (DAPI). Each meiotic prophase I substage is shown. The graph shows the number of MEILB2 foci associated with the chromosome axes. The mean value is shown as a red bar. *n* shows the analyzed oocyte number pooled from three mice. Scale bar, 5 μm. **b** Zygotene oocytes from WT (+/+) and *Meilb2* KO (−/−) females stained with the indicated antibodies and DAPI. Scale bar, 5 μm. **c-e** Zygotene oocytes from E14.5 WT (+/+) and *Meilb2* KO (−/−) females stained with SYCP3 in red, DAPI in blue, and RPA2 (**c**), DMC1 (**d**), or RAD51 (**e**) in green. The images in the other stages are shown in Supplementary Fig. 7a. The graph shows the number of RPA2 (**c**), DMC1 (**d**), or RAD51 (**e**) foci associated with the chromosome axes. The mean value is shown as a red bar. *n* shows the analyzed oocyte number pooled from two mice for each genotype. Scale bars, 5 μm. **f** Ovary sections from PD25 WT (+/+) and *Meilb2* KO (−/−) female mice stained with hematoxylin and eosin. The representative images of a primordial follicle (PF), primary follicle (PriF), and growing follicle (GF) are magnified. The graph shows the number of follicles in each ovary. The mean values of three independent experiments from three different mice are shown. Error bars show SD. Scale bars, 250 and 50 μm (magnified panel). **g** The average number of pups per litter. PD60 male (♂) and female (♀) pairs of WT (+/+) and *Meilb2* KO (−/−) genotypes were paired for >60 days of continuous breeding. *n* indicates the number of mating pairs examined. All analyses used two-tailed *t* tests. *N.S.* not significant. *$p < 0.05$, **$p < 0.01$, ****$p < 0.0001$. Source data are provided as a Source Data file

be explained by the presence of some redundant mechanisms targeting BRCA2 to the DSBs, such as the PALB2-mediated pathway reported in mitotic HR[50]. Collectively, the requirement of MEILB2 during meiotic HR for the recruitment of recombinases is a sexually conserved aspect of meiosis, but the degree of requirement is sexually dimorphic, similar to the other meiotic HR-related genes.

Given that *Brca2* is a well-established cancer-suppressor gene, we can speculate on the potential contribution of *Meilb2* to cancer development. Our RT-PCR experiment detected *Meilb2* mRNA expression only in germline tissues in mice suggesting that *Meilb2* does not function in normal somatic tissues. However, the publicly available cancer cell line databases, such as the Cancer Cell Line Encyclopedia[51], show that *MEILB2* mRNA is aberrantly upregulated in a number of human cancer cell lines, including breast and ovarian cancer cell lines (Supplementary Fig. 9). Furthermore, an aberrant fusion transcript of the *MEILB2* and *ZFP345* genes was discovered in a primary breast tumor sample from a human patient[52]. We argue for the possibility that the ectopic expression of MEILB2, or its fusion protein, in cancer cells can perturb the intrinsic BRCA2 function through direct binding, thus contributing to cancer development by disturbing the mitotic HR pathway. It will be an exciting challenge for future studies to investigate the function of MEILB2, as well as its fusion protein, in cancer cells and to investigate MEILB2 as a potential target for cancer therapy.

## Methods

**Mice**. KO mice for *Spo11* and *Dmc1* were reported earlier[18,53]. *Meilb2* KO mice were generated from an embryonic stem cell line, *Hsf2bp*tm1b(EUCOMM)Hmgu, purchased from the EMMA repository. The *Meilb2* allele was genotyped using the following primers: Common-Forward; 5′-GAG GAA GTG TTA CGT CGT CAC TAC-3′, WT-Reverse; 5′-CAC CAA CTG GCA GAC TGA CTC AAT-3′, KO-Reverse; 5′-CCT TCC TCC TAC ATA GTT GGC AGT-3′. All WT and KO mice were congenic with the C57BL/6J background. Animal experiments were approved by the Institutional Animal Care and Use Committee (#1316/18).

**Histological analysis**. Testes, epididymis, and ovaries were fixed in Bouin's fixative for 24 h at room temperature and embedded into paraffin blocks. Slices of 8-μm thickness were stained with hematoxylin and eosin. TUNEL analysis was carried out with an ApopTag Plus In Situ Apoptosis Fluorescein Detection Kit (S 7111; Millipore).

**Antibodies**. The following antibodies were used: rabbit antibodies against MEILB2 (this study) 1:1000, GFP (Invitrogen; A11122) 1:500, γH2AX (Abcam; ab11174) 1:2000, DMC1 (Santa Cruz Biotechnology; sc-22768) 1:500, RAD51 (Thermo Fisher Scientific; PA5-27195) 1:500, SPATA22 (Proteintech Group Inc; 16989-1-AP) 1:100, and SYCE3[54] 1:500; mouse antibodies against MEILB2 (this study) 1:500, DMC1 (this study) 1:2000, β-actin (Sigma; A2228-100UL) 1:2000, MLH1 (BD Biosciences; 51-1327GR) 1:50, and MYC (MBL; M192-3) 1:1000; rat antibody against RPA2 (Cell Signaling Technology; 2208) 1:500; sheep antibody against BRCA2[34] 1:50; chicken antibody against SYCP3 (this study) 1:3000; guinea pig antibody against histone H1T[36] 1:3000; and human anti-centromere antibody (Antibodies Incorporated; 15-234-0001) 1:100.

**Antibody production**. cDNAs encoding full-length *Dmc1*, *Sycp3*, and the N-terminus of *Meilb2* (a.a. 1–200) were cloned into the pET28c+ vector (Millipore). The HIS-tagged recombinant proteins were expressed in BL21 (DE3) cells, solubilized in a denaturing buffer (6 M HCl-guanidine and 30 mM Tris-HCl (pH 7.5)), and purified with Ni-nitrilotriacetic acid (QIAGEN). The recombinant proteins were dialyzed in phosphate-buffered saline (PBS) and used to immunize the animals. The polyclonal antibody against MEILB2 was affinity purified on antigen-coupled Sepharose beads (GE Healthcare).

**Reverse transcription PCR**. Total RNA was isolated from tissues using the RNeasy Mini Kit (Qiagen). cDNAs were generated by iScript™ reverse transcription super mix (Bio-Rad), and PCR amplification was performed using standard DNA polymerase. The primers used were as follows: *Meilb2*-forward; 5′-GCC TGC CGG AAC ATG GA-3′, *Meilb2*-reverse; 5′′-TGG TTT TGA CGA CCT CCT CG-3′, *Gapdh*-forward; 5′-TTC ACC ACC ATG GAG AAG GC-3′, and *Gapdh*-reverse; 5′-GGC ATG GAC TGT GTG GTC ATG A-3′.

**Exogenous protein expression in the testis**. Plasmid DNAs, pCAG-GFP-*Meilb2*, pCAG-GFP-*Brca2*-N (a.a. 1-981), pCAG-GFP-*Brca2*-M (a.a. 982-2035), pCAG-GFP-*Brca2*-C (a.a. 2117–2339), pCAG-GFP-*Brca2*-C ΔMBD (a.a. 2340-3329), and pCAG-GFP-*Brca2*-MBD (a.a. 2036–3329), were electroporated into live mouse testes as previously described[29]. Briefly, male mice at PD16–30 were anesthetized with pentobarbital, and the testes were pulled from the abdominal cavity. Plasmid DNA (10 μl of 5 μg/μl solution) was injected into each testis using glass capillaries under a stereomicroscope (M165C; Leica). Testes were held between a pair of tweezers-type electrodes (CUY21; BEX), and electric pulses were applied four times and again four times in the reverse direction at 35 V for 50 ms for each pulse. The testes were then returned to the abdominal cavity, and the abdominal wall and skin were closed with sutures. The testes were removed 24 h after the electroporation, and immunostaining was performed.

**Immunostaining of spermatocytes**. Testis cell suspensions were prepared in PBS, washed several times in PBS, and suspended in hypotonic buffer (30 mM Tris (pH 7.5), 17 mM trisodium citrate, 5 mM EDTA, and 50 mM sucrose) followed by suspension in 200 mM sucrose for mild spreads or in 100 mM sucrose for harsh spreads. After hypotonic treatment, the cell suspensions were placed on slides in the same volume of fixation buffer (1% paraformaldehyde and 0.1% Triton X-100 in PBS), fixed for 3 h at room temperature, and air-dried. For immunostaining, the slides were incubated with primary antibodies in PBS containing 5% bovine serum albumin (BSA) for 2 h and then with Alexa Fluor 488-, 594-, or 647-conjugated secondary antibodies (1:1000 dilution, Invitrogen) for 1 h at room temperature. The slides were washed with PBS and mounted with VECTASHIELD medium with 4,6-diamidino-2-phenylindole (DAPI; Vector Laboratories).

**Preparation of testis extract and immunoprecipitation**. Testes were removed from male C57BL/6J mice and suspended in extraction buffer (20 mM Tris-HCl (pH7.5), 50 mM KCl, 0.4 mM EDTA, 5 mM $MgCl_2$, 10% glycerol, 0.1% Triton X-100, and 1 mM β-mercaptoethanol) supplemented with cOmplete Protease Inhibitor (Roche) and Phosphatase Inhibitor (Roche). After homogenization, the cell extract was centrifuged at $50,000 \times g$ for 30 min at 4 °C and the supernatant was isolated. The extract was supplemented with Dynabeads protein A (Thermo Fisher Scientific) conjugated with 80 μg of anti-MEILB2 antibody or control IgG as the negative control and incubated for 6 h at 4 °C. The beads were washed with high-salt buffer (20 mM HEPES (pH 7.0), 400 mM KCl, 5 mM $MgCl_2$, 10% glycerol, 0.1% Triton X-100, and 1 mM β-mercaptoethanol) supplemented with cOmplete Protease Inhibitor (Roche) and Phosphatase Inhibitor (Roche). The samples were eluted with 0.1 M glycine (pH 2.5).

**Microscopy**. Images were obtained on a microscope (Olympus IL-X71 Delta Vision; Applied Precision) equipped with ×100 NA 1.40 and ×60 NA 1.42

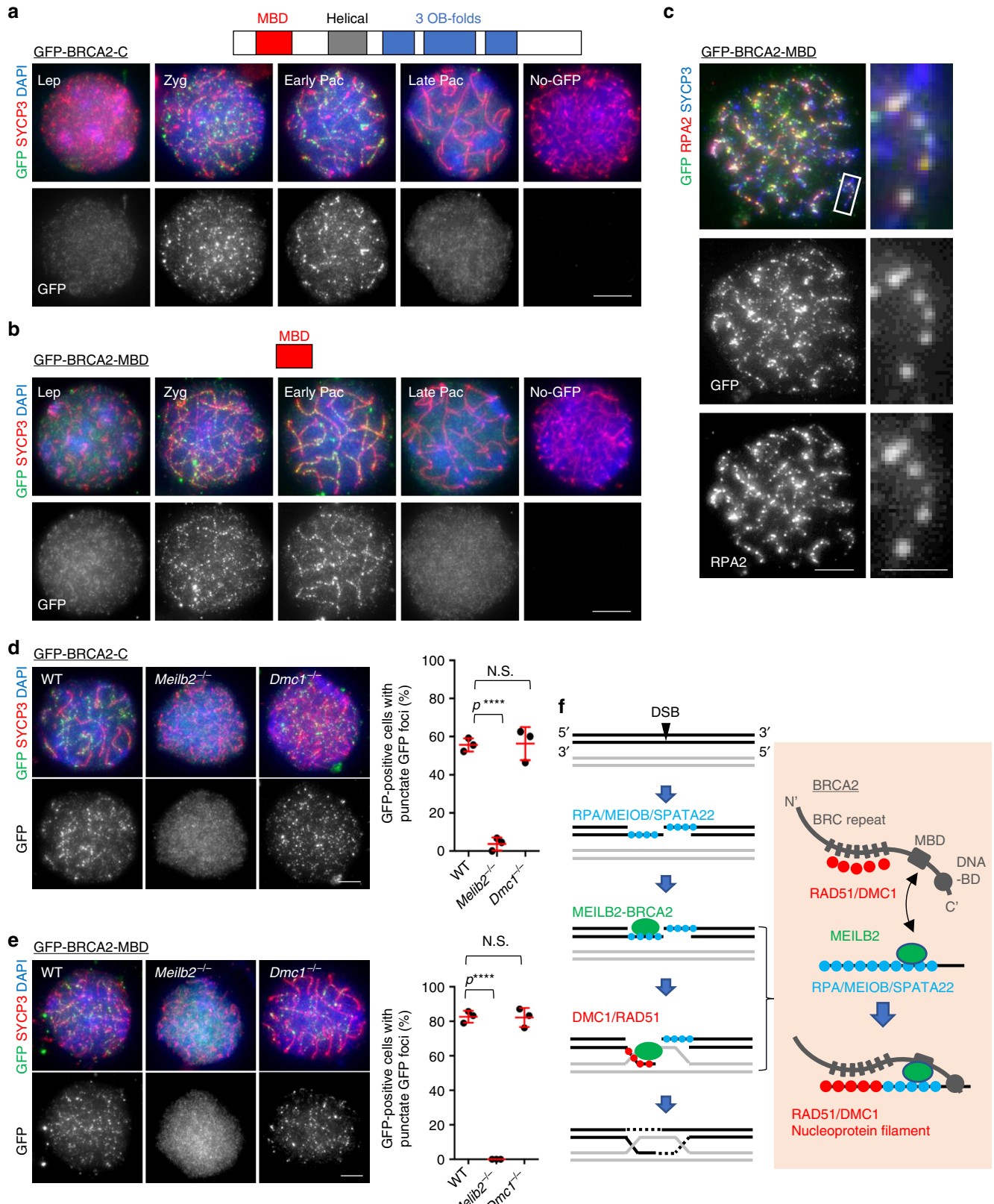

objectives, a camera (CoolSNAP HQ; Photometrics), and softWoRx 5.5.5 acquisition software (Delta Vision). Acquired images were processed with Photoshop (Adobe).

**Yeast two-hybrid assay**. Yeast two-hybrid screening was performed by Hybrigenics Services, Paris, France. The coding sequence for *Meilb2* was cloned into

pB27 as a C-terminal fusion to LexA (LexA-*Meilb2*). The construct was used as a bait to screen a random-primed mouse testis cDNA library constructed in pP6. Using a mating approach with YHGX13 and L40ΔGal4 yeast strains, 75 million clones (7-fold the complexity of the library) were screened. A total of 102 positive colonies were selected on a medium lacking tryptophan, leucine, and histidine and supplemented with 50 mM 3-aminotriazole. The prey fragments of the positive clones were amplified by PCR and sequenced at their 5′ and 3′ junctions. The

**Fig. 6** BRCA2 localization during meiotic prophase I. **a**, **b** Wild-type (WT) spermatocytes expressing GFP-BRCA2-C (**a**) or GFP-BRCA2-MBD (**b**) stained with the indicated antibodies and 4,6-diamidino-2-phenylindole (DAPI). **c** WT zygotene spermatocytes expressing GFP-BRCA2-MBD stained with the indicated antibodies and DAPI. The quantification of co-localization was performed using nine zygotene cells pooled from three electroporated mice. The axis-associated foci were counted. **d**, **e** Zygotene spermatocytes from WT, *Meilb2*−/−, and *Dmc1*−/− males expressing GFP-BRCA2-C (**d**) or GFP-BRCA2-MBD (**e**) stained with the indicated antibodies and DAPI. The graph shows the frequency of the green fluorescent protein (GFP)-positive zygotene spermatocytes with punctate GFP foci. The mean values of three independent experiments from three different electroporated mice are shown (GFP-BRCA2-C: 63 cells in WT, 57 cells in *Meilb2*−/−, and 85 cells in *Dmc1*−/−. GFP-BRCA2-MBD: 51 cells in WT, 53 cells in *Meilb2*−/−, and 76 cells in *Dmc1*−/−). Error bars show SD. **f** Schematic of the hierarchical loading of meiotic double-strand break (DSB)-associating proteins. After DSB formation by SPO11, the double-strand DNA is resected into single-strand DNA (ssDNA). RPA is loaded onto the ssDNA, which is epistatic to the localization of MEILB2. MEILB2 is then recruited to the DSBs, likely through binding to RPA or SPATA22-MEIOB, and this in turn recruits BRCA2 through the MEILB2–MEILB2-binding domain (MBD) interaction. BRCA2 then facilitates the loading of the RAD51 and DMC1 recombinases onto the ssDNA, resulting in the formation of a nucleoprotein filament that promotes DNA strand invasion. *Lep* leptotene, *Zyg* zygotene, *Pac* pachytene, *Dip* diplotene. All analyses used two-tailed *t* tests. N.S. not significant. ****$p < 0.0001$. Scale bars, 5 and 1 μm (magnified panel). Source data are provided as a Source Data file

resulting sequences were used to identify the corresponding interacting proteins in the GenBank database (NCBI) using a fully automated procedure.

For the yeast two-hybrid assay, *Meilb2*, *Dmc1*, *Rad51*, *Brca2*-MBD (a.a. 2117–2339), and *Brca2*-C (a.a. 2036–3329) cDNAs were cloned into the pGBKT7 vector. *Meilb2* cDNA was cloned into the pGADT7 vector. These bait and prey were co-transformed into the yeast strain AH109, and the positive transformants were selected on nutrition-restricted plates (SD-tryptophan-leucine-histidine-adenine).

**MI oocyte spreading**. Three-to-four-week-old female mice (WT and *Meilb2*−/−) were injected with 5 IU pregnant mare's serum gonadotropin intraperitoneally and were euthanized 44 h later. Their ovaries were chopped separately in M2 medium containing dbcAMP (100 μg/ml), and oocytes at the GV stage were collected and cultured in M16 medium under an oil drop at 37 °C and 5% $CO_2$. Oocytes were harvested when they reached the MI stage, and the zona pellucida was removed by acidic Tyrode's solution (Sigma-Aldrich). The oocytes were then fixed on glass slides with a drop of a solution containing 1% paraformaldehyde in distilled $H_2O$ (pH 9.2) and 0.3% Triton X-100. After air drying, the spreads were blocked with 3% BSA in PBS for 1 h at room temperature and incubated with primary antibodies at 4 °C overnight. After three washes, the slides were incubated with secondary antibodies (1:500 dilution) for 1 h at room temperature and the slides were mounted with VECTASHIELD medium with DAPI (Vector Laboratories).

**Follicle counting**. Ovaries from PD25 female mice (WT and *Meilb2*−/−) were fixed in 4% paraformaldehyde, dehydrated, and embedded in paraffin. Paraffin-embedded ovaries were then cut into 8-μm serial sections and stained with hematoxylin and eosin. The follicles were classified into three stages (primordial, primary, and growing follicles) and counted from the middle continuous sections based on Pedersen and Peters' standards[45].

**Cell culture**. Cell lines B16-F1 (Sigma) and C2C12 (Sigma) were maintained in Dulbecco's modified Eagle's medium (GIBCO Life Technologies) supplemented with 10% fetal bovine serum (Invitrogen), 100 U/ml Penicillin-Streptomycin (GIBCO Life Technologies), and 2.5 μg/ml Plasmocin (InvivoGen) in a humidified atmosphere of 5% $CO_2$ at 37 °C. Transfection was performed using Lipofectamine 2000 transfection reagent (Invitrogen) and Optimem (GIBCO Life Technologies).

**Pull-down assay**. Transfected B16-F1 cells were suspended in extraction buffer (20 mM Tris-HCl (pH7.5), 50 mM KCl, 0.4 mM EDTA, 5 mM $MgCl_2$, 10% glycerol, 0.1% Triton X-100, and 1 mM β-mercaptoethanol) supplemented with complete Protease Inhibitor (Roche) and Phosphatase Inhibitor (Roche). After sonication, the cell extract was centrifuged at $15,000 \times g$ for 30 min at 4 °C and the supernatant was isolated. The supernatant was then incubated with GFP-trap®_MA beads (ChromoTek) for 2 h at 4 °C on a rotating wheel. Beads were washed with high-salt buffer (20 mM HEPES (pH 7.0), 400 mM KCl, 5 mM $MgCl_2$, 10% glycerol, 0.1% Triton X-100, and 1 mM β-mercaptoethanol) supplemented with cOmplete Protease Inhibitor (Roche) and Phosphatase Inhibitor (Roche). The samples were eluted with sodium dodecyl sulfate loading buffer at 95 °C for 5 min.

**Quantification and statistical analysis**. The experiments were not randomized, so no statistical method was used to predetermine sample size, and the investigators were not blinded to allocation during the experiments or to outcome assessment. Each conclusion in the manuscript was based on results that were reproduced in at least two independent experiments and in at least two mice of each genotype. Sample sizes, statistical tests, and *p* values are indicated in the text, figures, and figure legends.

**Reporting Summary**. Further information on experimental design is available in the Nature Research Reporting Summary linked to this Article.

## Data availability

The data supporting the findings of this study are available from the corresponding author upon reasonable request. The source data underlying Figs. 1d–f, 3c, e, g–i, 4c–e, g, h, 5a, c–g, 6c–e and Supplementary Figs. 2b, 4a, c, 5b, c, 6, 7b and 8c are provided as a Source Data file.

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

## Acknowledgements

We thank Misun Kwon and Hyunsook Lee (Seoul National University) for providing the BRCA2 antibody, Mary Ann Handel (the Jackson Laboratory) for providing the histone H1T antibody, Meriem Echbarthi and Julie Grantham (University of Gothenburg) for sharing materials and valuable discussions, and Jayakrishnan Nandakumar (University of Michigan) for valuable discussions. This work was supported by Assar Gabrielsson's Foundation FB 17-10 (to H.S.) and O. E. och Edla Johanssons vetenskapliga stiftelse EK-f339/18 (to H.S.).

## Author contributions

J.Z. performed the experiments, J.Z. and H.S. analyzed the data, Y.F. and S.Y. contributed to the initial identification of *Meilb2*, and H.S. supervised the project and wrote the manuscript.

## Additional information

**Competing interests:** The authors declare no competing interests.

