## [Peer Review File · Nature Communications]

Reviewers' comments:

Reviewer #1 (Remarks to the Author):

Homology-directed repair of meiotic double-strand breaks (DSBs) necessitates the recruitment and assembly of strand-exchange proteins. Mechanisms that regulate this repair step are poorly understood, however, particularly in mammals. In this study, the authors identify a previously uncharacterized regulator of meiotic recombinases, MEILB2. They show that mice lacking MEILB2 fail to localize RAD51 and DMC1 recombinases, and as a result fail to complete meiotic recombination, leading to infertility in males and sub-fertility in females. A key finding is that MEILB2 interacts with a known regulator of recombinases, BRCA2, and is required for accumulation of GFP-BRCA2 at meiotic DSB sites in the form of cytologically observable complexes (foci).

This work highlights how DNA repair mechanisms are modulated in meiosis compared to mitosis and will be of interest to a broad audience. Few comments that could improve this study are listed below.

Major points:

1. The authors document the proportion of MEILB2 foci that colocalize with RPA2 and DMC1 foci (text related to Fig. 1), as well as the proportion of GFP-BRCA2-MBD foci that colocalize with RPA2 foci (text related to Fig. 6). The high proportion of colocalization of MEILB2 and GFP-BRCA2-MBD with RPA2 is inferred as association with DSB sites. These analyses should be elaborated on. How many cells were counted and from how many mice? Were only axis-associated foci counted? In the case of GFP-BRCA2-MBD, which cell stages were included in the quantitation? In most cases, it would be useful to state the colocalization proportions for both proteins in the pair being analyzed and not just for a single protein. Moreover, the colocalization analyses would be more convincing if the authors select few representative cells and state the proportion of axis-associated foci that colocalize when images of the individual proteins/fluorescence channels are offset or rotated ninety degrees with respect to one another. This is especially important given the representative images shown are of spermatocytes that do not appear to be spread extensively, thus increasing the likelihood that a large proportion of foci will colocalize by chance.

2. The authors examine SPATA22 localization in mice lacking MEILB2 (Supplementary Fig. 3b) and conclude that SPATA22 loading is similar to wild type. This point is also raised in the discussion section and the authors propose that SPATA22-MEIOB may be the scaffold for MEILB2 on ssDNA. A comparison of SPATA22 foci numbers during leptotene is not useful as majority of spermatocytes classified as leptotene by the authors have negligible foci. And the data for zygotene cells show significantly elevated SPATA22 foci numbers in mice lacking MEILB2. Does this still hold true if more spermatocytes are counted (more than 16 cells). If so, its implications should be discussed. Elevated SPATA22 foci numbers may result from an accumulation of incomplete repair intermediates in the absence of MEILB2 and is intriguing given that RPA2 foci numbers remain unchanged.

3. The authors examine the colocalization of GFP-BRCA2-MBD and RPA2 foci (Fig. 6b,c) and conclude that the MBD region is sufficient for GFP-BRCA2 localization. They also discuss that the BRCA2-MBD-MEILB2 interaction is the major pathway for recruiting BRCA2 to meiotic DSB sites, rather than BRCA2-ssDNA interactions. For this conclusion, it is important to determine whether the number of foci formed and/or the number of foci that colocalize with RPA2 are the same between GFP-BRCA2-C and GFP-BRCA2-MBD. In addition, while the colocalization of GFP-BRCA2 and MEILB2 is implied by their mutual colocalization with RPA2, it would be useful to definitively know that this is the case and know the extent of their colocalization.

4. The phenotype reported here resembles that described for mice lacking TEX15 (retention of RPA

foci, lack of DMC1 foci, male-specific infertility; Yang et al., JCB, 2008) and should be discussed, or at least mentioned.

Additional minor points:

5. Although the number of RPA2 foci detected in the absence of MEILB2 are similar to wild type (Fig. 4c), it is unlikely that these foci reflect accumulation at similar recombination structures. While RPA2 foci in wild type would include accumulation at recombination intermediates formed after strand invasion, those formed in the absence of MEILB2 and strand invasion are expected to be comprised of RPA2 that has not yet been displaced by DMC1/RAD51. The authors state that the similar numbers of RPA2 foci in wild type and knockout animals shows that DSB formation and resection proceed normally. While the data are consistent with this interpretation and the gamma-H2AX signal accumulation in knockouts, the authors should discuss that the pattern of RPA2 foci in wild type and knockouts may reflect distinct RPA2 behaviors.

6. Images of TUNEL-stained testis sections (Fig. 3e) are difficult to see. Authors should enhance visibility of the DAPI channel.

7. The authors should confirm that DMC1 is still expressed in the absence of MEILB2 by Western blotting.

8. The authors present a schematic of the hierarchical loading of meiotic recombination proteins in Fig. 4g. Although the data presented do suggest that MEILB2 associates with axes after DSB formation and likely prior to DMC1 filament formation, the depiction of MEILB2 as an ssDNA binding protein is misleading and should be modified. The same schematic is reproduced in Fig. 6f, however here it is accompanied by an inset that clarifies this.

9. For plots depicting quantitation of foci number, the authors state the number of cells analyzed (n). It would be useful to also state the number of mice analyzed, either in figure legends or methods.

10. The following errors require correction:

- Page 8 line 156; pull-downed should be pulled-down.
- Supplementary Fig. 2 legend; Statement "each meiotic ..." should be removed.
- Supplementary Fig. 3 graph; late pac cells should be included in wild-type group.
- Fig. 5 c,d,e and Fig. 6f legends have typos.

Reviewer #2 (Remarks to the Author):

Review for NCOMMS-18-23721-T

In this manuscript submission Jingjing Zhang, Yasuhiro Fujiwara, Shohei Yamamoto and Hiroki Shibuya present exciting work detailing a novel interaction partner of BRCA2, they call MEILB2, which is involved in homologous DNA recombination processes during meiosis, upstream of RAD51 and DMC1 strand invasion. The authors use a series of experiments that include in vivo electroporation technique to visualize MEILB2, immunoprecipitation, Y2H, immunofluorescence microscopy, mouse mutant analysis, and, combine in vivo electroporation and mouse mutant analysis together to provide convincing evidence of a link between early meiotic recombination events and later ones that involve MEILB2 and BRCA2.

Comments:

1. The authors describe the localization of MEILB2 using chromatin spreads, and their manuscript demonstrates the link between MEILB2-BRCA2 and RAD51/DMC1. RAD51 and DMC1 remain late on the X-Y chromosome as inter-sister repair is predominant on these chromosomes. I would like

to see the authors present are more detailed analysis of the localization pattern of MEILB2 in this context. Does it also persist on the X-Y?

2. Immunoprecipitation studies – I think it would be beneficial to extend the IP assays a little. Probing for SPATA22-MEIOB and RPA proteins for example.

3. I also think including localization of MEIOB and SPATA22 data within the main content of the manuscript is important. These two also link early and late stages of HR during meiosis, and they present some supplemental data. However, based on the discussion, and overlap with function and timing, more data should be provided within the main data presented. For example, could there be something very interesting to observe with regards to the sexually dimorphic Meilb2 KO phenotype? Perhaps there is compensation by MEIOB-SPATA22.

4. The graph in Supplemental Figure 3 has an error with regards to the bars used to separate the data for each genotype (i.e. there are 4 stages shown for control, and only two for mutant, not 3 and 3).

Reviewer #3 (Remarks to the Author):

In this manuscript, the authors utilize their previously reported *in vivo* electroporation technique to uncover a key germ cell-specific protein they term MEILB2 that localizes to chromosomal axes and facilitates the repair of meiotic DSBs. The authors report that in male mice, loss of Meilb2 leads to nearly complete impairment of DMC1 and RAD51 localization to meiotic DSBs, resulting in failed meiotic DSB repair and sterility. Interestingly, the authors find the effect of Meilb2 loss to be sexually dimorphic, resulting in only partial loss of recombinase localization and subfertility in female mice. Further, the authors demonstrate that MEILB2 interacts directly with a specific region of BRCA2, the termed the MEILB2 Binding Domain (MBD), and is necessary for the localization of exogenous GFP-tagged BRCA2 fragments to the chromosomal axis preceding DSB repair.

This work provides evidence for a new component, MEILB2, of the meiotic DSB repair machinery in which MEILB2 localizes BRCA2, and hence its associated recombinases, DMC1 and RAD51, to programmed DSBs. Given its critical role in males and supportive role in females, uncovering MEILB2 is an important finding and will be of interest to many in the meiosis field as well as those studying DSB repair and BRCA2. Interesting approaches not standard to the meiosis field were used in this study, including identification of MEILB2 through a screen of upregulated genes in germ cells and testing them via testis electroporation for a characteristic localization pattern. The experiments are mostly compelling, although often descriptive since quantification is frequently lacking. Moreover, although it may be difficult in this manuscript to uncover the biochemical mechanism by which MEILB2 acts, greater insight could be obtained by additional experiments.

Major points:

1. Why does MEILB2 “linger” on chromosome axes longer than RAD51 and DMC1? This suggests an additional role for MEILB2 unrelated to BRCA2 and recombinase loading. Since MEILB2 co-localizes with ssDNA binding proteins, does MEILB2 have later functions in meiosis, as does RPA, such that focusing on the interaction with BRCA2 is too limited? Does MEILB2 interact with RPA?

2. The description of the MEILB2 domain structure is limited. How many ARM repeats are there in the domain? Is the ARM domain where the MBD of BRCA2 binds? What other proteins interact with MEILB2 that would provide insight into its mechanism of recruiting BRCA2? The authors state that there are no sequence-predicted DNA binding domains, but have they tested this *in vitro*?

3. It is surprising that the MEILB2 IP in Fig 2e pulls down RAD51 but not DMC1, at least not very

robustly. Several questions are raised by this result. If DMC1 were co-expressed in Fig. 1c along with the other proteins, would it be pulled down by BRCA2-M as is RAD51? There is a non-essential DMC1 binding site at BRCA2 F2406; could binding of MEILB2 so close by (2117-2339) abrogate binding of DMC1? Reciprocal IP experiments to that in Fig. 1e would further shed light on this discrepancy.

4. Reduced RAD51 and DMC1 foci in the MEILB2 mutant suggests a defect in loading these proteins to DSB sites. However, the trivial explanation that expression of these proteins is reduced should be addressed by western blotting in males and RT-PCR in females.

5. It is true that sexually dimorphic phenotypes have been reported for several meiotic mutants, but generally not core HR proteins. Thus, it is still surprising that MEILB2 loss would have such a mild effect on females if it is a core HR. The discussion should expand on this point more, even if more insight into the dimorphism cannot be provided. Moreover, there is no discussion or data with regards to PALB2, which is also supposed to be a critical localizer of BRCA2. Could PALB2 be the critical localizer of BRCA2 in females, explaining why MEILB2 is not as critical as it is in males?

6. The primary data need to be provided, including the number of mice used for each experiment

Additional points in order of appearance:

Line 55: "During meiotic prophase I, HR takes place using not only sister chromatids..". What is the evidence that sister chromatids are an important repair template during meiosis? The focus is usually on homologous chromosomes. (Indeed this contradicts a statement on line 70.)

Line 105: How many and which candidate genes were tested in the in vivo electroporation experiments? This should be provided as a supplemental table.

Line 115: Is MEILB2 present in non-vertebrate species, especially model organisms used to study meiosis? MEILB2 appears to be much more highly conserved than BRCA2. Is the MBD of BRCA2 well conserved? A sequence should be provided in a supplemental figure.

Fig. 1c: The tissues-specific expression in Fig. 1c suggests that only germ cells express MEILB2, but it is mentioned later that some cancer cell lines aberrantly express MEILB2. To firm up the tissue-specificity, it would be advisable to examine expression in embryos and/or MEFs.

Fig. 1d: It is difficult to determine if the staging is always correct. For example in Fig 1d, all of the chromosomes do not appear to be completely synapsed at the image labeled "early pachynema".

Line 122: western blot analysis should be provided to determine the specificity of the MEILB2 antibody.

Line 129: MEILB2 is said to colocalize with RPA2, but the limited number of foci shown suggests that foci may be adjacent rather than perfectly co-localization. How many RPA2 foci colocalized with MEILB2?

Line 138: Colocalization of MEILB2 and RAD51 needs to be quantified.

Line 144: What other proteins were found to interact with MEILB2 in a Y2H screen? This should be provided as a list with the number of times each protein was found.

Fig. 3c: Please provide data on testis weights from several mice. Also, it is important to show juvenile mouse testis weights to verify the small testis phenotype is due solely to defects after meiotic entry.

Fig. 3d: the images are too small and not of high enough quality. A more intensive assessment of

the histology should be given.

Fig. 3f: Clearly sperm are substantially reduced but again the images are too small for clarity. Given the staging in 3h, it seems unlikely that any sperm will be formed so this figure would be better placed in the supplement.

Fig. 3g: the mutant and control cells are at different stages.

Fig. 3h: it's not clear if the cells are in leptotema, since some synapsis may have started. Are fully synapsed homologs ever apparent? A block at pachynema seems fairly clear, but H1t staining would determine whether any cells have progressed to mid-pachynema.

Fig. 3i: How often do partner switches observed? Quantification should be provided.

Fig. 4a,b: gH2AX signal needs to be quantified at early zygonema.

Line 210: SPATA22 staining is supposed to be similar, but it actually the foci counts are higher, which might be expected if resected intermediates cannot load RAD51 and DMC1.

Line 225: If spermatocytes do not progress to pachynema, MLH1 crossovers will not form, so this sentence needs to be modified. That is, the lack of MLH1 foci doesn't say anything about whether meiotic HR is totally abolished if the cells do not progress to a stage when MLH1 foci form.

Line 280: section on "MEILB2 recruits BRCA2 to the meiotic recombination sites". Experiments are performed with BRCA2 truncations via an electroporation technique looking at GFP fusions. They certainly support this conclusion but should be stated with some caution, given their indirectness. Is there no experimental setup that circumvents the need for in vitro electroporation of these fragments, and instead uses endogenous BRCA2, like chromatin-bound fractionation and probing for levels by Western blot?

Fig. 4f: quantification of MEILB2 foci in Dmc1 mice would be beneficial.

Fig. 4g: left panel is unnecessary and repetitive with the closing model figure.

Fig. 6a: Localization of BRCA2-C- Δ MBD needs to be presented. Moreover, localization of BRCA2-N and M should be determined as well to determine if the MBD is essential for localization and domain mapping should be similarly provided for MEILB2.

Line 329: Y2H may not be sufficient to detect interactions. Unless direct experiments are shown, the HSF2 argument is weak.

Line 381: It is stated that females don't require BRCA2, based on a transgenic mouse study (ref 27). This is highly misleading, unless it was definitively shown in this reference that the transgenic females are not expressing BRCA2 at all during meiotic prophase.

Line 387: "argue" should be replaced by "speculate" since there is no data presented. Moreover, all of the experiments are performed in mouse cells and the only expression analysis is RT-PCR of several tissues, so it is not certain that there is no mitotic expression in some human tissues.

We thank the Editor and Reviewers for providing constructive comments on our manuscript. Please find below our point-by-point responses to all the reviewer's comments.

Reviewer #1

1. The authors document the proportion of MEILB2 foci that colocalize with RPA2 and DMC1 foci (text related to Fig. 1), as well as the proportion of GFP-BRCA2-MBD foci that colocalize with RPA2 foci (text related to Fig. 6). The high proportion of colocalization of MEILB2 and GFP-BRCA2-MBD with RPA2 is inferred as association with DSB sites. These analyses should be elaborated on. How many cells were counted and from how many mice? Were only axis-associated foci counted? In the case of GFP-BRCA2-MBD, which cell stages were included in the quantitation? In most cases, it would be useful to state the colocalization proportions for both proteins in the pair being analyzed and not just for a single protein. Moreover, the colocalization analyses would be more convincing if the authors select few representative cells and state the proportion of axis-associated foci that colocalize when images of the individual proteins/fluorescence channels are offset or rotated ninety degrees with respect to one another. This is especially important given the representative images shown are of spermatocytes that do not appear to be spread extensively, thus increasing the likelihood that a large proportion of foci will colocalize by chance.

We have repeated the colocalization analysis of MEILB2 and RPA2/DMC1 with the following changes.

- 1, We have quantified the colocalization proportions for both proteins in the pair being analyzed.
- 2, We clearly mentioned the number of cells and number of mice used for the quantifications in the figure legend.
- 3, We have mentioned that "the axis-associated foci were counted". See below.

Results line 130-135

"The majority of the MEILB2 foci co-localized with the ssDNA-binding protein RPA2, which marks the site of resected DNA at the DSB sites, suggesting that MEILB2 foci largely corresponded to DSB sites (Fig. 1e; 75% and 84% of the RPA2 foci stained positive for MEILB2 and 85% and 85% of the MEILB2 foci stained positive for RPA2 at the late leptotene and zygotene stages, respectively)."

Results line 139-142

"most of the DMC1 foci accompanied MEILB2 foci (Fig. 1f; 65% and 81% of the DMC1 foci stained positive for MEILB2 and 17% and 31% of the MEILB2 foci stained positive for DMC1 at the late leptotene and zygotene stages, respectively)."

Figure legend Fig.1e, f

"The quantification of co-localization was performed using three late leptotene cells and six zygotene cells pooled from two mice for RPA2 (e) and six late leptotene cells and ten zygotene cells pooled from three mice for DMC1 (f). The axis-associated foci were counted."

According to the suggestion by reviewer #3, we have also quantified the colocalization of MEILB2 and RAD51 using the same method as followed (Supplementary Fig. 2b).

Results line 146-149

“We also confirmed the partial co-localization of MEILB2 and RAD51 (Supplementary Fig. 2b; 68% and 76% of the RAD51 foci stained positive for MEILB2 and 29% and 42% of the MEILB2 foci stained positive for RAD51 at the late leptotene and zygotene stages, respectively)”

Figure legend in supplementary Fig.2b

“The quantification of co-localization was performed using four late leptotene cells and eight zygotene cells pooled from two mice. The axis-associated foci were counted.”

2. The authors examine SPATA22 localization in mice lacking MEILB2 (Supplementary Fig. 3b) and conclude that SPATA22 loading is similar to wild type. This point is also raised in the discussion section and the authors propose that SPATA22-MEIOB may be the scaffold for MEILB2 on ssDNA. A comparison of SPATA22 foci numbers during leptotene is not useful as majority of spermatocytes classified as leptotene by the authors have negligible foci. And the data for zygotene cells show significantly elevated SPATA22 foci numbers in mice lacking MEILB2. Does this still hold true if more spermatocytes are counted (more than 16 cells).

We have increased the cell number and confirmed that the elevated SPATA22 foci number in *Meilb2* KO zygotene spermatocytes is statistically significant (Fig. 4g).

If so, its implications should be discussed. Elevated SPATA22 foci numbers may result from an accumulation of incomplete repair intermediates in the absence of MEILB2 and is intriguing given that RPA2 foci numbers remain unchanged.

We have quantified the signal intensity of SPATA22 foci and found that it significantly increased in *Meilb2* KO zygotene spermatocytes compared to the WT, suggesting that SPATA22 (both foci number and signal intensity) is somehow stabilized in *Meilb2* KO spermatocytes, as you pointed out (Supplementary Fig. 5c).

We have added these new data with the interpretations as followed.

Results line 243-250

“The staining of SPATA22, which associates with the recombination intermediates by forming a complex with the meiosis-specific ssDNA-binding protein MEIOB, showed that the loading of SPATA22 occurred normally in the leptotene stage but the foci abnormally accumulated toward the zygotene stages in *Meilb2* $-/-$ spermatocytes compared to the WT (Fig. 4g). Further, the signal intensity of SPATA22 in *Meilb2* $-/-$ zygotene spermatocytes was significantly higher than WT (Supplementary Fig. 5c). These data suggest that the incompletely repaired

recombination intermediates, which are associated with SPATA22-MEIOB, accumulated in *Meilb2*^{-/-} spermatocytes.”

3. The authors examine the colocalization of GFP-BRCA2-MBD and RPA2 foci (Fig. 6b,c) and conclude that the MBD region is sufficient for GFP-BRCA2 localization. They also discuss that the BRCA2-MBD-MEILB2 interaction is the major pathway for recruiting BRCA2 to meiotic DSB sites, rather than BRCA2-ssDNA interactions. For this conclusion, it is important to determine whether the number of foci formed and/or the number of foci that colocalize with RPA2 are the same between GFP-BRCA2-C and GFP-BRCA2-MBD. In addition, while the colocalization of GFP-BRCA2 and MEILB2 is implied by their mutual colocalization with RPA2, it would be useful to definitively know that this is the case and know the extent of their colocalization.

We have conducted the colocalization analyses of GFP-BRCA2-C/HBD and RPA2. As a result, both GFP-BRCA2-C and HBD colocalized with RPA2 in an almost comparable manner (Fig. 6c and Supplementary Fig. 8c).

Results line 331-335

“the majority of GFP-BRCA2 signals co-localized with the endogenous RPA2 signals in the zygote stage (Fig. 6c and Supplementary Fig. 8c; 73% and 68% of GFP foci were stained for RPA2 and 80% and 61% of RPA2 foci were stained for GFP in GFP-BRCA2-MBD and GFP-BRCA2-C expressing cells, respectively)”

Figure legend Fig.6c

“The quantification of co-localization was performed using nine zygote cells pooled from three electroperated mice. The axis-associated foci were counted.”

Figure legend Supplementary Fig.8c

“The quantification of co-localization was performed using nine zygote cells pooled from three electroperated mice. The axis-associated foci were counted.”

4. The phenotype reported here resembles that described for mice lacking *TEX15* (retention of RPA foci, lack of *DMC1* foci, male-specific infertility; Yang et al., *JCB*, 2008) and should be discussed, or at least mentioned.

The molecular characters of TEX15, such as the localization pattern of TEX15 or its interacting proteins, were poorly defined in the previous study and, therefore, it is difficult to get functional insights into this protein and to discuss its function relationship between MEILB2. However, we agree that the *Tex15*KO phenotype was similar to *Meilb2* KO mice. We have mentioned this point in the discussion part as followed.

Discussion line 363-366

“The knockout mice of *Tex15*, a poorly characterized meiotic gene, also showed similar phenotypes in male meiosis, implying the potential functional interplay between TEX15 and MEILB2-BRCA2 complex.”

5. Although the number of RPA2 foci detected in the absence of MEILB2 are similar to wild type (Fig. 4c), it is unlikely that these foci reflect accumulation at similar recombination structures. While RPA2 foci in wild type would include accumulation at recombination intermediates formed after strand invasion, those formed in the absence of MEILB2 and strand invasion are expected to be comprised of RPA2 that has not yet been displaced by DMC1/RAD51. The authors state that the similar numbers of RPA2 foci in wild type and knockout animals shows that DSB formation and resection proceed normally. While the data are consistent with this interpretation and the gamma-H2AX signal accumulation in knockouts, the authors should discuss that the pattern of RPA2 foci in wild type and knockouts may reflect distinct RPA2 behaviors.

We agree that it is important point to mention. We have added the discussion as followed.

Discussion line 367-372

“It is known that RPA complex remains on meiotic recombination sites even after the removal of recombinases until the pachytene stage, suggesting that RPA complex somehow associates with joint molecules at this stage. Therefore, even though the number of RPA foci is comparable between WT and *Meilb2* KO zygotene spermatocytes, these RPA foci likely represent different recombination intermediates, i.e. both unrepaired DSBs and joint molecules in WT and unrepaired DSBs in *Meilb2* KO, respectively.”

6. Images of TUNEL-stained testis sections (Fig. 3e) are difficult to see. Authors should enhance visibility of the DAPI channel.

We have enhanced visibility of the DAPI channel.

7. The authors should confirm that DMC1 is still expressed in the absence of MEILB2 by Western blotting.

We have confirmed by Western Blot that DMC1 protein is comparably expressed as WT in the absence of MEILB2 (Fig. 4f). We have added the interpretation as well.

Results line 237-239

“We confirmed that the protein expression of DMC1 and RAD51 was comparable between WT and *Meilb2*^{-/-}, proving that MEILB2 is needed for the localization, but not expression, of recombinases (Fig. 4f)”

8. The authors present a schematic of the hierarchical loading of meiotic recombination proteins in Fig. 4g. Although the data presented do suggest that MEILB2 associates with axes after DSB formation and likely prior to DMC1 filament formation, the depiction of MEILB2 as an ssDNA binding protein is misleading and should be modified. The same schematic is reproduced in Fig. 6f, however here it is accompanied by an inset that clarifies this.

According to the suggestion from reviewer#3, we have deleted the whole Fig.4g schematic. Also, we have changed the Fig.6f schematic in order to make it clear that MEILB2 is not a ssDNA-binding protein.

9. For plots depicting quantitation of foci number, the authors state the number of cells analyzed (n). It would be useful to also state the number of mice analyzed, either in figure legends or methods.

We have added the analyzed cell number and mouse number for every quantification data either in main text or figure legends.

10. The following errors require correction:

-Page 8 line 156; pull-downed should be pulled-down.

-Supplementary Fig. 2 legend; Statement “each meiotic ...” should be removed.

-Supplementary Fig. 3 graph; late pac cells should be included in wild-type group.

-Fig. 5 c,d,e and Fig. 6f legends have typos.

We have corrected these errors.

Reviewer #2

1. The authors describe the localization of MEILB2 using chromatin spreads, and their manuscript demonstrates the link between MEILB2-BRCA2 and RAD51/DMC1. RAD51 and DMC1 remain late on the X-Y chromosome as inter-sister repair is predominant on these chromosomes. I would like to see the authors present more detailed analysis of the localization pattern of MEILB2 in this context. Does it also persist on the X-Y?

We agree that it is interesting point and we have added the data, where we stained the early pachytene spermatocytes with DMC1, MEILB2 and SYCP3 (Supplementary Fig. 2a). As you pointed out, DMC1 foci were restricted to the sex chromosomes in this early pachytene cell. However, MEILB2 foci were still abundant even along autosomes and never restricted on sex chromosomes. These data further support the notion that MEILB2 retained on the recombination nodules even after the DMC1 removal. We have added the interpretation as well.

Results line 142-146

“in the early pachytene stage, the DMC1 foci mostly disappeared from autosomes and became restricted to the sex chromosomes (Supplementary Fig. 2a), while MEILB2 foci were still abundant even along autosomes, supporting the notion that MEILB2 remained on the recombination nodules even after the removal of DMC1 (Supplementary Fig. 2a).”

2. Immunoprecipitation studies – I think it would be beneficial to extend the IP assays a little. Probing for SPATA22-MEIOB and RPA proteins for example.

We have repeated MEILB2 IP and blotted with RPA1, RPA2 and SPATA22. In this assay, we have detected the interaction with SPATA22 but not with RPA1 or RPA2 (Supplementary Fig. 5d).

Accordingly, we have added the interpretation as followed in the results section.

Results line 250-253

“We also detected the in vivo interaction between MEILB2 and SPATA22, but not the RPA complex, by MEILB2 immunoprecipitation implying some potential functional interplay between MEILB2 and SPATA22 (Supplementary Fig. 5d).”

3. I also think including localization of MEIOB and SPATA22 data within the main content of the manuscript is important. These two also link early and late stages of HR during meiosis, and they present some supplemental data. However, based on the discussion, and overlap with function and timing, more data should be provided within the main data presented.

We have repeated the quantification of SPATA22 foci number in *Meilb2* KO spermatocytes and added the data in main figure (Fig. 4g).

Also, we have quantified the signal intensity of SPATA22 and found that SPATA22 foci became around 3 times stronger in *Meilb2*KO zygote spermatocytes compared to the WT (Supplementary Fig. 5c).

For example, could there be something very interesting to observe with regards to the sexually dimorphic *Meilb2* KO phenotype? Perhaps there is compensation by MEIOB-SPATA22.

Because SPATA22 foci rather increased in *Meilb2*KO spermatocytes, where the stronger phenotype was detected compared to the oocytes, we are thinking the compensation by MEIOB-SPATA22 may not explain the sexual dimorphic phenotypes of *Meilb2*KO mice.

Instead, we have added several new interpretations and discussions regarding these newly arising issues in result part of the manuscript as followed.

Results line 243-250

“The staining of SPATA22, which associates with the recombination intermediates by forming a complex with the meiosis-specific ssDNA-binding protein MEIOB, showed that the loading of SPATA22 occurred normally in the leptotene stage but the foci abnormally accumulated toward the zygotene stages in *Meilb2*^{-/-} spermatocytes compared to the WT (Fig. 4g). Further, the signal intensity of SPATA22 in *Meilb2*^{-/-} zygotene spermatocytes was significantly higher than WT (Supplementary Fig. 5c). These data suggest that the incompletely repaired recombination intermediates, which are associated with SPATA22-MEIOB, accumulated in *Meilb2*^{-/-} spermatocytes.”

Discussion line 422-424

“The milder defects in our *Meilb2* KO female can be explained by the presence of some redundant mechanisms targeting BRCA2 to the DSBs, such as the PALB2-mediated pathway reported in mitotic HR.”

4. The graph in Supplemental Figure 3 has an error with regards to the bars used to separate the data for each genotype (i.e. there are 4 stages shown for control, and only two for mutant, not 3 and 3).

We have corrected the error.

Reviewer #3

1. Why does MEILB2 “linger” on chromosome axes longer than RAD51 and DMC1? This suggests an additional role for MEILB2 unrelated to BRCA2 and recombinase loading. Since MEILB2 co-localizes with ssDNA binding proteins, does MEILB2 have later functions in meiosis, as does RPA, such that focusing on the interaction with BRCA2 is too limited?

We agree that is important point we should mention in the paper. We added the following sentences in the discussion part.

Discussion line 372-376

“similar to the RPA case, a significant number of MEILB2 foci remained on the recombination intermediates until early pachytene stages in WT meocytes implying the possibility that MEILB2 also binds to joint molecules and could have some additional functions in the later stage of prophase I, such as the stabilization of joint molecules.”

Does MEILB2 interact with RPA?

We have repeated MEILB2 immunoprecipitation and showed that MEILB2 does not interact with RPA complex (RPA1 and RPA2) (Supplementary Fig. 5d). However, MEILB2 interacts with SPATA22 suggesting that

MEILB2 could have functional interplay with SPATA22-MEIOB complex. We have added these discussions in the result section.

Results line 250-253

“We also detected the *in vivo* interaction between MEILB2 and SPATA22, but not the RPA complex, by MEILB2 immunoprecipitation implying some potential functional interplay between MEILB2 and SPATA22 (Supplementary Fig. 5d).”

2. The description of the MEILB2 domain structure is limited. How many ARM repeats are there in the domain? The predicted 3D structure of MEILB2 drawn by Phyre2 program is attached (image colored by rainbow N to C terminus). From this prediction, we can see 4 ARM repeats (each ARM repeat is defined to have around 40 a.a. and consists of 2-3 helices that curves to form a hairpin like structure).

Is the ARM domain where the MBD of BRCA2 binds?

We have not analyzed the domain-specific function or protein interaction of MEILB2 yet. We are now trying to solve the crystal structure of MEILB2 as well as MEILB2-BRCA2 complex to answer the question.

What other proteins interact with MEILB2 that would provide insight into its mechanism of recruiting BRCA2?

We have provided the new data suggesting that MEILB2 also forms complex with SPATA22 *in vivo* (Supplementary Fig. 5d). Indeed, SPATA22 is somehow stabilized in *Meilb2*KO spermatocytes, suggesting some potential functional interplay between MEILB2 and SPATA22.

I agree that the identification and analysis of additional interactors of MEILB2 will provide more insight into MEILB2-BRCA2 functions. To this end, we are now screening for novel binding proteins of MEILB2, by MEILB2 immunoprecipitation and mass spectrometry, and will report the results in follow-up studies in the future.

The authors state that there are no sequence-predicted DNA binding domains, but have they tested this *in vitro*? We are now purifying recombinant MEILB2 protein and MEILB2-BRCA2 complex for their biochemical characterizations, such as testing their DNA binding activities and mediator activities for recombinases, as well as for protein crystallization. We will report these follow-up studies that focus on the biochemical aspects of MEILB2-BRCA2 complex in the future studies.

3. It is surprising that the MEILB2 IP in Fig 2e pulls down RAD51 but not DMC1, at least not very robustly. Several questions are raised by this result. If DMC1 were co-expressed in Fig. 1c along with the other proteins, would it be pulled down by BRCA2-M as is RAD51? There is a non-essential DMC1 binding site at BRCA2 F2406; could binding of MEILB2 so close by (2117-2339) abrogate binding of DMC1? Reciprocal IP experiments to that in Fig. 1e would further shed light on this discrepancy.

The interaction between BRCA2 and DMC1 is reported using *in vitro* systems and it is still controversial whether BRCA2 and DMC1 interact *in vivo* or not.

We are now making BRCA2 polyclonal antibody to address this point, and will immunoprecipitate endogenous BRCA2 complex from testis extracts to test whether BRCA2 interacts with DMC1 or not. We will report these results in the future studies.

4. Reduced RAD51 and DMC1 foci in the MEILB2 mutant suggests a defect in loading these proteins to DSB sites. However, the trivial explanation that expression of these proteins is reduced should be addressed by western blotting in males and RT-PCR in females.

We have confirmed by Western Blot that DMC1 protein is comparably expressed as WT in the absence of MEILB2 in male (Fig. 4f). We have added the interpretation as well.

Results line 237-239

“We confirmed that the protein expression of DMC1 and RAD51 was comparable between WT and *Meilb2*^{-/-}, proving that MEILB2 is needed for the localization, but not expression, of recombinases (Fig. 4f)”

5. It is true that sexually dimorphic phenotypes have been reported for several meiotic mutants, but generally not core HR proteins. Thus, it is still surprising that MEILB2 loss would have such a mild effect on females if it is a core HR. The discussion should expand on this point more, even if more insight into the dimorphism cannot be provided. Moreover, there is no discussion or data with regards to PALB2, which is also supposed to be a critical localizer of BRCA2. Could PALB2 be the critical localizer of BRCA2 in females, explaining why MEILB2 is not as critical as it is in males?

We agree that the sexually dimorphic phenotype of our *Meilb2* KO mice is interesting. We added the discussion describing the possibility that PALB2 mediated BRCA2-targeting mechanism can bypass the requirement of MEILB2 in female meiosis.

Discussion line 422-424

“The milder defects in our *Meilb2* KO female can be explained by the presence of some redundant mechanisms targeting BRCA2 to the DSBs, such as the PALB2-mediated pathway reported in mitotic HR.”

6. The primary data need to be provided, including the number of mice used for each experiment

We have provided all the primary data for every quantification data in the corresponding figure legends, figure or manuscript.

Line 55: “During meiotic prophase I, HR takes place using not only sister chromatids..”. What is the evidence that sister chromatids are an important repair template during meiosis? The focus is usually on homologous chromosomes. (Indeed this contradicts a statement on line 70.)

The partner choice for DSB repair, either inter homologous chromosome (IH) or inter sister chromatid (IS), have been investigated in budding yeast model systems, where the ration of IH:IS was estimated to be 5:1 (Kim KP, Weiner BM, Zhang L, Jordan A, Dekker J, Kleckner N. Sister cohesion and structural axis components mediate homolog bias of meiotic recombination. Cell 143, 924-937 (2010).). We have cited this paper and changed the manuscript as below.

Before: “During meiotic prophase I, HR takes place using not only sister chromatids, but also homologous chromosomes as a repair template, resulting in the formation of crossover structures between homologous chromosomes.”

After: “During meiotic prophase I, HR takes place using homologous chromosomes as the primary repair template rather than sister chromatids, resulting in the formation of crossover structures between homologous chromosomes.”

Line 105: How many and which candidate genes were tested in the in vivo electroporation experiments? This should be provided as a supplemental table.

The list of genes upregulated in embryonic ovary is shown in the reference paper (Soh YQ, Junker JP, Gill ME, Mueller JL, van Oudenaarden A, Page DC. A Gene Regulatory Program for Meiotic Prophase in the Fetal Ovary. PLoS Genet 11, e1005531 (2015).). We have found HSF2BP/MEILB2 from this list and studied its function in this paper, as described in line 103-110.

We are still in the process of examining the localization of the other candidate meiotic genes and would like to keep the gene of interest confidential at this point.

Line 115: Is MEILB2 present in non-vertebrate species, especially model organisms used to study meiosis? MEILB2 appears to be much more highly conserved than BRCA2.

We only found MEILB2 homologs in vertebrate species (Supplementary fig.1) and could not find MEILB2 homologs out of vertebrate species, similar to the BRCA2 case.

Is the MBD of BRCA2 well conserved? An sequence should be provided in a supplemental figure.

We have provided the sequence alignment of BRCA2 MBD domains in vertebrate species in supplementary fig.3. Indeed, MBD is highly conserved between species (62% identity between human and mouse, while 56% identity in the case of full-length BRCA2 between human and mouse).

```
H.sapiens      -OLVLGTVKSVLVENIHVLCKEOASPKNVKMEIGKTEFFSDVVPVKTNIEVCSTYSK-----
P.troglodytes -OLVLGTVKSVLVENIHVLCKEOASPKNVKMEIGKTEFFSDVVPVKTNIEVCSTYSK-----
M.mulatta     -OLVLGTVKSVLVENIHVLCKEOASPKNVKMEIGKTEFFSDVVPVKTNIEVCSTYSK-----
C.lupus      -OLLVGSKGSVLVENIHVLCKEOALPKNIKTEIGKAEFFPNLPVKTNIEFCSTYSK-----
B.taurus     -OSVLGTRVSHSTDNHLLGKQOTLPKYIKKEIGKTEFFPDL-VKINTEICSTDSK-----
M.musculus   -TOLVLGTVKSVSHS-KANLLGKQOTLPQNIKVKTDKMTFFSDVVPVKTNV--GEYYSK-----
R.norvegicus -TOSVLGTVKVSOR-KTNILKQKONLQNIKIESNKMTEFFSDVSMKTNV--GEYYSK-----
G.gallus     -SAPFKNSFEQE-ETRFFRNG-ELNLGIR-----AESESDL-----CSATSKAEINI
```

```
H.sapiens      --DSENYFETEAVEIAKAFMEDDELTDSELPKSHATHSLFTCPENEEMVLSNSRIKRRGE
P.troglodytes --DSENYFETEAVEIAKAFMEDDELTDSELPKSHATHSLFTCPENEEMVLSNSRIKRRGE
M.mulatta     --DSENYFETEAVEIAKAFMEDGELTDSELPKSHATHSLFTCPONEEMVLSNSRIKRRGE
C.lupus      --DPENYFETEAVEIAKAFMEDGELTDSELLSHAKHVFVFCQNTKEMVLSNSRIKRRGD
B.taurus     --DPENYFETEAVEIAKAFMEDGELTDSEFPKSHAKHSPVFCQKNEETVLSNSRIKRRGD
M.musculus   --ESENIFETEAVEIAKAFMEDDELTDSEQ-THAKCSLFTCPONEET--LNSRTRKRRGM
R.norvegicus --EPENYFETEAVEIAKAFMEDDELTDSEQ-THAKCSLFTCPONEA--LNSRTRKRRGM
G.gallus     -FQTPKDYLLTEAVEIAKAFMEDD-LSDSGVQVKSQSPFGKMSD-----FQNKPFKRRHL
```

```
H.sapiens      PLILVGEPSIKRNLLNEFDRIIENQEKSLKASKSTPDGTIKDRRLFMHHVSLPNTCVPF
P.troglodytes PLILVGEPSIKRNLLNEFDRIIENQEKSLKASKSTPDGTIKDRRLFMHHVSLPNTCVPF
M.mulatta     ALISAGEPPIKRNLLNEFDRIIENQEKSLKASKSTPDGTIKDRRLFMHHVSLPNTCVPF
C.lupus      ALVSVGEPPKRNLLNEFDRIIKKQETS LKASKSTPDGIIKDRSLFMHHVSLPNTCVPF
B.taurus     ALVTVGEPPKRNLLNEFDRIIENQEKSLKASKSTPDGAMKDRRLFMHHVSLPNTCVPF
M.musculus   TVDAVGOPPIKRNLLNEFDRIIESKGS LTPSKSTPDGTVKDRSLFTHHSLPNTCVPF
R.norvegicus AGVAVGOPPIKRNLLNEFDRIIESKGS LTPSKSTPDGTIKDRRLFTHHSLPNTCVPF
G.gallus     EKDSHGEPPKRNLLNEFEKX-KIPPKSVKPLKSTPDGIFKDRRLFMHVVPLKPVTCRPL
```

```
H.sapiens      RTTKEROEIQPNPFPAPGOEF---LSKSHLYEHLTLEKSSSNLAVSGHFFYQVSATRNE
P.troglodytes RTTKEROEIQPNPFPAPGOEF---LSKSHLYEHLTLEKSSSNLAVSGHFFYQVSATRNE
M.mulatta     CTTTKEROEIQPNPFPAPGOEF---LSKSHLYEHLTLEKSSSNLAVSGHFFYQVSATRNE
C.lupus      RTTKEROEIQPNPFPAPGOEF---LPKSHLYEHLTLEKSSSNLAVSRQPFVCMVPATGNE
B.taurus     CTTTKEROEIQPNPFPAPGOEF---LSKSHLYEHLTLEKSSSNLAVSGHFFYQVSATRNE
M.musculus   CSSKERQGAORPHLTSAPQEL---LSKSHLYEHLTLEKSSSNLAVSRQPFVCMVPATGNE
R.norvegicus CSSKERQETQSPHVSAPQGL---QSKGHPWRHSALEKSSSNPIVSI LPAHDVSATRTE
G.gallus     GTTKEROEVRNPTLALPDQDLKGFKSI PAVFOHCALRQSSSGASGLFTPHK-AVAKDSE
```

Fig. 1c: The tissues-specific expression in Fig. 1c suggests that only germ cells express MEILB2, but it is mentioned later that some cancer cell lines aberrantly express MEILB2. To firm up the tissue-specificity, it would be advisable to examine expression in embryos and/or MEFs.

Fig. 1d: It is difficult to determine if the staging is always correct. For example in Fig 1d, all of the chromosomes do not appear to be completely synapsed at the image labeled “early pachynema”.

We have changed the early pachytene picture, where we can see the complete synapsis along autosomes.

Line 122: western blot analysis should be provided to determine the specificity of the MEILB2 antibody.

We have provided the western blot results using our MEILB2 antibody in Fig.3b and Fig.4f, where we can find a specific band at the estimated molecular weight from mouse testis extract that totally disappeared in *Meilb2* KO testis sample. This data proved that our MEILB2 antibody was specifically detecting endogenous MEILB2 protein from testis extract in this western blot experiment.

Line 129: MEILB2 is said to colocalize with RPA2, but the limited number of foci shown suggests that foci may be adjacent rather than perfectly co-localization. How many RPA2 foci colocalized with MEILB2?

We have quantified the co-localization of MEILB2 and RPA2 foci more carefully. As a result, respectively, 75% and 84% of RPA2 foci were stained for MEILB2 and 85% and 85% of MEILB2 foci were stained for RPA2 at the late leptotene and zygotene stages. We have described these data as followed in the main text.

Results line 130-135

“The majority of the MEILB2 foci co-localized with the ssDNA-binding protein RPA2, which marks the site of resected DNA at the DSB sites, suggesting that MEILB2 foci largely corresponded to DSB sites (Fig. 1e; 75%

and 84% of the RPA2 foci stained positive for MEILB2 and 85% and 85% of the MEILB2 foci stained positive for RPA2 at the late leptotene and zygotene stages, respectively).”

Line 138: Colocalization of MEILB2 and RAD51 needs to be quantified.

We have quantified the co-localization of MEILB2 and RAD51 foci more carefully and put a new representative picture (Supplementary fig.2b). As a result, respectively, 68% and 76% of RAD51 foci were stained for MEILB2 and 29% and 42% of MEILB2 foci were stained for RAD51 at the late leptotene and zygotene stages. We have described these data as followed in the main text.

Results line 146-150

“We also confirmed the partial co-localization of MEILB2 and RAD51 (Supplementary Fig. 2b; 68% and 76% of the RAD51 foci stained positive for MEILB2 and 29% and 42% of the MEILB2 foci stained positive for RAD51 at the late leptotene and zygotene stages, respectively), which was consistent with the notion that DMC1 and RAD51 co-localize with each other.”

Line 144: What other proteins were found to interact with MEILB2 in a Y2H screen? This should be provided as a list with the number of times each protein was found.

We have identified 19 genes, including BRCA2, as a potential interactor of MEILB2 in the Y2H screening. We would like to keep their gene names confidential at this point, because we are now analyzing some of the candidate genes for the future studies. We will provide the whole list in the follow-up studies.

Fig. 3c: Please provide data on testis weights from several mice.

We have provided data on testis weights from 3 adult mice (6 testes in total) for WT and KO in Fig.3c with quantification and statistical analyses.

Also, it is important to show juvenile mouse testis weights to verify the small testis phenotype is due solely to defects after meiotic entry.

We have provided data on testis weights from 2 juvenile mice (4 testes in total) for WT and KO in Supplementary Fig. 4a with quantification and statistical analyses. As a result, there was no significant difference in their testis weight, suggesting that the defects occur after meiotic entry. We have described these data in the main text.

Results line 187-189

“The juvenile testes at PD14 showed no size difference between *Meilb2* $^{-/-}$ and WT, suggesting that the defects likely occur after meiotic entry (Supplementary Fig. 4a).”

Fig. 3d: the images are too small and not of high enough quality. A more intensive assessment of the histology should be given.

The submitted PDF file was compressed to meet the file size limit for the initial submission and the resolution of some images was not so high enough. We will submit the original high-resolution images later. Also, we have enlarged the images and highlighted spermatids by arrowheads in the Fig. 3d and added more detailed description in the figure legend as followed.

Figure legend Fig. 3d

“Testis sections from 8-week-old WT ($+/+$) and *Meilb2* KO ($-/-$) male mice stained with hematoxylin and eosin. The arrowheads indicate spermatids, which are present in WT ($+/+$) but not in *Meilb2* KO ($-/-$) testis sections.”

Fig. 3f: Clearly sperm are substantially reduced but again the images are too small for clarity. Given the staging in 3h, it seems unlikely that any sperm will be formed so this figure would be better placed in the supplement.

We have enlarged the images and put it in supplementary fig.4b.

Fig. 3g: the mutant and control cells are at different stages.

We have put another *Meilb2* KO cell as a representative picture, that is in the mid-zygotene stage and we can see several synapsis as the WT zygotene cell control.

Fig. 3h: it's not clear if the cells are in leptotema, since some synapsis may have started.

We have described the cell staging criteria in the figure legend of Fig.3g. In short, we have categorized SYCP3-positive early prophase I cells which **lacks any SYCE3 signal** (synapsis marker) along chromosome axes into the leptotene stage. These criteria are standard criteria accepted in the preceding studies.

We have intensified the SYCE3 signals of the leptotene cells from WT and *Meilb2*KO in the below pictures.

Even in these intensified pictures, we cannot see any SYCE3 stretch along chromosome axes, so that these cells are surely in the leptotene stage.

Are fully synapsed homologs ever apparent? A block at pachynema seems fairly clear, but H1t staining would determine whether any cells have progressed to mid-pachynema.

We have described the cell staging criteria in the figure legend of Fig.3g and fully synapsed cells are categorized into the pachytene stage. Based on this criterial, we performed the quantification shown in Fig.3g graph. As a result, pachytene cells were totally absent in *Meilb2*KO testes.

We also added the histone H1T staining and quantification data in fig. 3h, confirming that H1T signal was totally absent in *Meilb2*KO spermatocytes.

Results line 200-203

“We further confirmed the cell cycle arrest at early prophase I stage in *Meilb2*^{-/-} spermatocytes by the absence of histone H1T staining (Fig. 3h), which starts to appear from the mid-pachytene stage in WT spermatocytes.”

Fig. 3i: How often do partner switches observed? Quantification should be provided.

We have quantified the frequency of partner switch. As a result, 92% of *Meilb2* KO zygotene spermatocytes showed at least one partner switch (25 cells pooled from three mice). We described this quantification in the figure legend of Fig. 3i as below.

Figure legend Fig. 3i

“(i) Zygotene spermatocytes from *Meilb2* KO (-/-) males stained with the indicated antibodies and DAPI. The partner-switched chromosomes are magnified to the right. About 92% of *Meilb2* KO zygotene spermatocytes showed at least one partner switch (25 zygotene cells pooled from three mice).”

Fig. 4a,b: gH2AX signal needs to be quantified at early zygonema.

We have quantified the frequency of gH2AX positive zygotene spermatocytes. As a result, the 100% of zygotene cells showed gH2AX signal both in WT and *Meilb2* KO cells (30 cells pooled from three mice for each genotypes). We have described this quantification in the figure legend of Fig. 4a,b as below.

Figure legend Fig.4a,b

“(a) Spermatocytes from WT (+/+) males stained with the indicated antibodies and DAPI. All of the WT zygotene nuclei showed cloudy γH2AX signals (30 cells pooled from three mice).

(b) Spermatocytes from *Meilb2* KO (-/-) males stained with the indicated antibodies and DAPI. All of the *Meilb2* KO zygotene nuclei showed cloudy γH2AX signals (30 cells pooled from three mice).”

Line 210: SPATA22 staining is supposed to be similar, but it actually the foci counts are higher, which might be expected if resected intermediates cannot load RAD51 and DMC1.

We have repeated the quantification of SPATA22 foci number in *Meilb2* KO spermatocytes and, indeed, SPATA22 foci number was increased in *Meilb2* KO spermatocytes. We have added the data (Fig. 4g) as well as the interpretations as below.

Results line 243-250

“The staining of SPATA22, which associates with the recombination intermediates by forming a complex with the meiosis-specific ssDNA-binding protein MEIOB, showed that the loading of SPATA22 occurred normally in the leptotene stage but the foci abnormally accumulated toward the zygotene stages in *Meilb2*^{-/-} spermatocytes compared to the WT (Fig. 4g). Further, the signal intensity of SPATA22 in *Meilb2*^{-/-} zygotene spermatocytes was significantly higher than WT (Supplementary Fig. 5c). These data suggest that the incompletely repaired recombination intermediates, which are associated with SPATA22-MEIOB, accumulated in *Meilb2*^{-/-} spermatocytes.”

Line 225: If spermatocytes do not progress to pachynema, MLH1 crossovers will not form, so this sentence needs to be modified. That is, the lack of MLH1 foci doesn't say anything about whether meiotic HR is totally abolished if the cells do not progress to a stage when MLH1 foci form.

We have changed the sentences as followed.

Before: “The staining of MLH1, a marker of sites that are destined to become crossovers, confirmed the total abolishment of crossover formation in *Meilb2*^{-/-} spermatocytes (Supplementary Fig. 4b), suggesting that meiotic HR is totally abolished in *Meilb2*^{-/-} spermatocytes likely because of the mislocalization of recombinases.”

After: “The staining of MLH1, a marker of sites that are destined to become crossovers, confirmed the total abolishment of crossover formation in *Meilb2*^{-/-} spermatocytes (Supplementary Fig. 5b), consistent with the cell cycle arrest at the zygotene stage due to the mislocalization of recombinases.”

Line 280: section on “MEILB2 recruits BRCA2 to the meiotic recombination sites”. Experiments are performed with BRCA2 truncations via an electroporation technique looking at GFP fusions. They certainly support this conclusion but should be stated with some caution, given their indirectness. Is there no experimental setup that circumvents the need for in vitro electroporation of these fragments, and instead uses endogenous BRCA2, like chromatin-bound fractionation and probing for levels by Western blot?

We have tried several different antibodies against mouse BRCA2 but all didn't work for IF in meocytes likely because of the limited protein amount of endogenous BRCA2 (one of the data is shown in Supplementary Fig. 8a). This is likely the reason why BRCA2 localization in meiosis has been controversial until today, even though it is evident from the phenotype assay that BRCA2 functions at meiotic DSB sites.

Our electroporation technique solved this technical barrier and for the first time managed to visualize the BRCA2 localization at meiotic DSB sites. We are thinking these localization data are convincing because the GFP signal is quite specific at meiotic DSBs in early prophase I, which never seen in non-electroporated cells (Fig.6a,b: most right panels).

Fig. 4f: quantification of MEILB2 foci in *Dmc1* mice would be beneficial.

We have quantified MEILB2 foci in zygotene spermatocytes from WT, *Dmc1* KO and *Spo11* KO in Fig. 4h.

Fig. 4g: left panel is unnecessary and repetitive with the closing model figure.

We have removed the whole Fig. 4g, because it was repetitive with the closing model as pointed out.

Fig. 6a: Localization of BRCA2-C-ΔMBD needs to be presented. Moreover, localization of BRCA2-N and M should be determined

We have added the localization analyses of BRCA2-C-ΔMBD, BRCA2-N and BRCA2-M in Supplementary Fig. 8b. These data reinforce our conclusion that MBD of BRCA2 is necessary and sufficient for the DSB localization.

Results line 322-324 and 326-329

“While we did not observe any specific localization of GFP-BRCA2-N or GFP-BRCA2-M (Supplementary Fig. 8b), we were able to detect the recombination nodule-like foci of GFP-BRCA2-C on the chromosome axes (Fig. 6a).”

“The GFP-BRCA2-C lacking the MBD (GFP-BRCA2-C ΔMBD) showed cloudy nuclear signals and failed to localize along chromosome axis (Supplementary Fig. 8b). Together these results suggest that the MBD is necessary and sufficient for the chromosome axis localization of BRCA2.”

as well to determine if the MBD is essential for localization and domain mapping should be similarly provided for MEILB2.

We are now trying to analyze the domain specific function of MEILBS by solving the crystal structure of MEILB2 as well as MEILB2-BRCA2 complex. We will report these follow-up studies in the future.

Line 329: Y2H may not be sufficient to detect interactions. Unless direct experiments are shown, the HSF2 argument is weak.

We agree and have deleted the following sentences.

“Because there were no follow-up studies confirming the MEILB2-HSF2 interaction under physiological conditions in the testis, and because our yeast two-hybrid screening failed to identify HSF2 as a MEILB2 interactor, we are skeptical of the significance of the MEILB2-HSF2 interaction under physiological conditions.”

Line 381: It is stated that females don't require BRCA2, based on a transgenic mouse study (ref 27). This is highly misleading, unless it was definitively shown in this reference that the transgenic females are not expressing BRCA2 at all during meiotic prophase.

We agree and have deleted the following sentences.

“This was also the case for the Brca2 KO mice carrying the human BRCA2 gene, where male mice showed complete sterility but females produced functional oocytes that could develop into embryos.”

Line 387: “argue” should be replaced by “speculate” since there is no data presented. Moreover, all of the experiments are performed in mouse cells and the only expression analysis is RT-PCR of several tissues, so it is not certain that there is no mitotic expression in some human tissues.

We have replaced “argue” into “speculate”.

[Redacted text block consisting of four horizontal black bars]

Reviewers' comments:

Reviewer #1 (Remarks to the Author):

The authors have addressed comments satisfactorily. The additional quantification analyses and SPATA22 data have improved what was already an intriguing and thorough study. I have no additional concerns.

Reviewer #2 (Remarks to the Author):

No further comments. Thank you for addressing my prior review comments.

Reviewer #3 (Remarks to the Author):

The authors have addressed many of the reviewers' concerns, although in some cases, the authors address the comments but do not update the manuscript.

Primary data should be provided in a data file.

Nomenclature comment: MEIOB is an established meiotic-specific protein. "MEILB2" sounds very much like MEIOB and is likely to cause some confusion.

Line 22:

RAD51/DMC1 foci are not abolished in females, so this line needs to be clarified.

Line 105:

Authors:

We are still in the process of examining the localization of the other candidate meiotic genes and would like to keep the gene of interest confidential at this point.

-The impact of the manuscript is somewhat diminished by not including this information.

Presumably, several tested candidates did not show an interesting localization pattern, yet readers will not know which ones these are and may do needless experiments.

Line 112:

Authors: The predicted 3D structure of MEILB2 drawn by Phyre2 program is attached (image colored by rainbow N to C

terminus). From this prediction, we can see 4 ARM repeats

-Is it a problem to mention 4 ARM repeats in the text? The analysis here seems superficial.

Line 115:

Authors: We only found MEILB2 homologs in vertebrate species (Supplementary fig.1) and could not find MEILB2

homologs out of vertebrate species, similar to the BRCA2 case.

-BRCA2 is found in several non-vertebrate species. The legend to Supplementary Fig. 1 is not very informative. If MEILB2 is not found in non-vertebrate species, this should be stated and commented upon with regards to the more widespread distribution of BRCA2.

Authors: We have provided data on testis weights from 3 adult mice (6 testes in total) for WT and KO in Fig.3c with

quantification and statistical analyses.

We have provided data on testis weights from 2 juvenile mice (4 testes in total) for WT and KO in Supplementary Fig. 4a with quantification and statistical analyses.

-It is unusual to provide weights for individual testes.

Authors: We have quantified the frequency of gH2AX positive zygotene spermatocytes.
-To quantify gH2AX, the fluorescence intensity needs to be measured.

Fig. 1d and throughout:

It is not clear what criteria were used for staging chromosome spreads when only SYCP3 staining is used. For example, the two Leptotene spermatocytes are very different in panels d and e and it's not clear that the early pachytene cell is instead a late zygotene cell.

Further, the authors switch colors in d and e for SYCP3. This is unnecessary.

Sup Fig. 3:

There are no landmarks for the sequences (i.e., what amino acids are being aligned).

Line 126:

A maximum value (327) is not very useful; the mean should be given.

Line 184:

Testes vs body weight should be reported, or at least body weight to show that the mice are similar in size.

Line 187:

The staining of Fig. 3d is very strong. Please point out which cells are considered to be spermatogonia and spermatocytes in the mutant.

Line 247:

It is likely that MEIOB accumulates in foci in the mutant but it is nevertheless not shown, so MEIOB should be removed from this line.

Line 271:

Variation can exist between animals at this stage. How many embryos were analyzed?

Supp Fig 7b: corrupted figure: Data not visible in the graph.

Line 305:

This sentence contradicts the next (or at least is confusing); from the cited paper it is clear that Brca2 nullizygosity leads to meiotic arrest in males and that Meilb2 mutant

Line 413:

The authors suggest that the complete loss of RAD51/DMC1 foci in males is due to a stronger checkpoint function in females. I am not aware of checkpoints regulating focus formation. This needs to be clarified.

Line 430:

The authors refer to the CCLE to say that Meilb2 unregulation is observed in cancers, but not analysis is provided for this point.

Fig. 6f. In the model, RPA recruits MEILB2. This is confusing since MEILB2 does not interact with RPA, and SPATA22, but not RPA, accumulate in Meilb2-/-.

Please find below our point-by-point responses to the reviewer #3's requests.

Reviewer #3

1. Primary data should be provided in a data file.

We have provided the primary data in the file "Source data.xlsx".

2. Nomenclature comment: MEIOB is an established meiotic-specific protein. "MEILB2" sounds very much like MEIOB and is likely to cause some confusion.

We are thinking that the name "MEILB2 (Meiotic localizer of BRCA2)" is appropriate. That is because 1, the name directly represents the molecular function and 2, it is a widely accepted nomenclature to put "MEI" on the top of the meiotic genes (even when they sound similar) as shown below.

MEIOB...Meiosis-specific with OB domain-containing protein

MEIOC...Meiosis-specific coiled-coil domain-containing protein

MEISETZ...Meiosis-induced factor containing PR/SET domain and zinc-finger motif

MEIKIN... Meiosis-specific kinetochore protein

3. Line 22: RAD51/DMC1 foci are not abolished in females, so this line needs to be clarified.

We appreciate your suggestion and clearly mentioned that the disruption of RAD51/DMC1 localization occurs in male (spermatocytes).

line 22-24

"Disruption of Meilb2 abolished the localization of RAD51 and DMC1 recombinases in spermatocytes, leading to errors in DSB repair and male sterility."

4. Line 105: The impact of the manuscript is somewhat diminished by not including this information.

Presumably, several tested candidates did not show an interesting localization pattern, yet readers will not know which ones these are and may do needless experiments.

We are thinking that it is acceptable not to show the localization of the other genes on this paper because 1, the localization of other genes is out of focus and not related to the characterization and functional analyses of MEILB2 (that is the main focus of our paper), and 2, we didn't examine the localization of other genes using a combination of various experimental approaches as we did for MEILB2 in this paper and we cannot show the data with confidence at this point. We would like to publish the other gene's data when the data become publication quality in the future.

5. Line 112: Is it a problem to mention 4 ARM repeats in the text? The analysis here seems superficial.

We appreciate your suggestion and have mentioned that ARM repeat domain of MEILB2 is composed of 4 ARM repeats.

line 111-113

"MEILB2 is a 338 amino acid protein composed of an N-terminus coiled-coil domain and a C-terminus armadillo repeat domain (composed of four armadillo repeats) (Fig. 1b)."

6. Line 115: BRCA2 is found in several non-vertebrate species. The legend to Supplementary Fig. 1 is not very informative. If MEILB2 is not found in non-vertebrate species, this should be stated and commented upon with regards to the more widespread distribution of BRCA2.

We appreciate your suggestion and have mentioned that BRCA2 is more widely conserved than MEILB2.

line 363-365

"While Brca2 is conserved in some invertebrate species such as nematode, MEILB2 homologs are found only in vertebrate species, suggesting that MEILB2 evolved later than BRCA2 and was specialized for the meiosis-specific regulation of BRCA2."

7. It is unusual to provide weights for individual testes.

We have provided the testes vs body weight ratio instead of testis weight in Fig. 3c.

8. To quantify γ H2AX, the fluorescence intensity needs to be measured.

We appreciate your suggestion and have quantified the signal intensity of γ H2AX in zygotene spermatocytes in Supplementary Fig. 4c.

9. Fig. 1d and throughout: It is not clear what criteria were used for staging chromosome spreads when only SYCP3 staining is used. For example, the two Leptotene spermatocytes are very different in panels d and e and it's not clear that the early pachytene cell is instead a late zygotene cell.

Further, the authors switch colors in d and e for SYCP3. This is unnecessary.

The staining of synapsis marker (such as central element protein SYCE3) will tell the strict substages of meiotic prophase I. We used these strict criteria to define the meiotic substage distribution in WT and *Meilb2* KO by the double staining of SYCP3 and SYCE3 (Fig. 3g).

However, when people count the foci number of recombination nodule proteins, it is more common method to define the meiotic prophase I substages using the staining of lateral element **alone** (such as SYCP3). The distribution of SYCP3 signal is enough to define the meiotic- prophase I substages, since SYCP3 shows distinct distribution pattern in each substage; dotted (early leptotene) or discontinuous thread (late leptotene), continuous thread with incomplete synapsis (zygotene), continuous thread with complete synapsis (pachytene), continuous thread with dis-synapsis (diplotene).

The below are example where people used SYCP3 staining alone to define the meiotic prophase I substages during the counting of recombination nodule proteins.

RNF212 foci counting in different substages

Fig.1 of this Dr. Neil Hunter lab paper (<https://www.nature.com/articles/ng.2858>).

MEIOB foci counting in different substages

Fig.2 of this Dr. Jeremy Wang lab paper (<https://www.nature.com/articles/ncomms3788>).

MEIOB foci counting in different substages

Fig.2 of this Dr. Gabriel Livera lab paper

(<https://journals.plos.org/plosgenetics/article?id=10.1371/journal.pgen.1003784>).

MEI4 foci counting in different substages

Fig. 4 of this Dr. Bernard de Massy paper (<http://genesdev.cshlp.org/content/24/12/1266.long>).

SPATA22 foci counting in different substages

Fig. 1 of this Dr. Kazuhiro Kitada paper (<https://www.nature.com/articles/srep06148>).

There are several reasons why double staining is preferred for these foci counting (for example; people sometimes do not have antibodies compatible for the triple staining), but the most critical reason is that immunostaining signals tend to be more leaky between different channels when we perform triple staining compared to the case of double staining. The leakage of the signal can disturb the observation and counting of faint dotted signals of meiotic recombination nodules. So that double staining was preferred in the most of the preceding studies for foci counting, and we followed this convention in our study (such as MEILB2 counting in Fig. 1d).

You also mentioned that “the two Leptotene spermatocytes are very different in panels d and e”. This is because SYCP3 signal in leptotene stage showed distinct pattern in early leptotene (dotted signal; Fig. 1d) and in late leptotene (discontinuous thread; Fig. 1e). We have specified this point in the figure legend of Fig. 1e, saying that the leptotene cells in Fig. 1e is “late leptotene cells”.

line 452-456

“(e, f) WT spermatocytes stained with the indicated antibodies and DAPI. The co-localizing foci along the chromosome axis are highlighted by yellow arrowheads. The quantification of co-localization was performed using three **late leptotene** cells and six zygotene cells pooled from two mice for RPA2 (e) and six **late leptotene** cells and ten zygotene cells pooled from three mice for DMC1 (f).”

You also questioned about the difference between zygotene and early pachytene cells. As I mentioned above zygotene and pachytene can be distinguished by the SYCP3 signals; continuous thread with incomplete synapsis (zygotene) and continuous thread with complete synapsis (pachytene). I have picked up the SYCP3 signal of Fig. 1d zygotene cells in the below pictures. You can see the clearly branched axis shown by arrow (left) or in the magnified picture (right), suggesting that homologous synapsis is not yet completed in this cell suggesting that this cell is in zygotene stage.

Furthermore, during the quantification throughout the paper, we have also used the original 3D pictures (before projection) and carefully observed these SYCP3 signals in order to define the substages.

You also mentioned that “Further, the authors switch colors in d and e for SYCP3. This is unnecessary”.

The purpose of Fig 1. d is to show the localization of MEILB2 with stage marker SYCP3, so that we showed these two in green and red respectively.

The purpose of Fig 1. e is to compare the localization of MEILB2 and RPA2 (using SYCP3 as a stage marker), so that we showed these two in green and red respectively.

Those are reasons why we switched the colour in Fig. 1d and 1e.

10. Sup Fig. 3: There are no landmarks for the sequences (i.e., what amino acids are being aligned).

We have described the amino acid number in the figure legend of Supplementary Fig. 3 as below.

Supplementary Fig. 3. Sequence alignment of MBD domains of BRCA2 in vertebrates.

Sequence data are from the NCBI protein database. *H. sapiens* (NP_000050.2, 2164-2391 a.a.), *P. troglodytes* (XP_509619.2, 2164-2391 a.a.), *M. mulatta* (XP_001118184.2, 2106-2333 a.a.), *C. lupus* (NP_001006654.2, 2181-2408 a.a.), *B. taurus* (XP_002691853.1, 2171-2397 a.a.), *M. musculus* (NP_033895.2, 2117-2339 a.a.), *R. norvegicus* (NP_113730.2, 2132-2354 a.a.), *G. gallus* (NP_989607.2, 2124-2339 a.a.)

11. Line 126: A maximum value (327) is not very useful; the mean should be given.

We showed the average value of foci instead of maximum.

line 125-126

“MEILB2 foci started to appear from the leptotene stage, reached their greatest number in the zygotene stage (257 foci on average)”

12. Line 184: Testes vs body weight should be reported, or at least body weight to show that the mice are similar in size.

We have provided the testes vs body weight ratio instead of testis weight.

13. Line 187: The staining of Fig. 3d is very strong. Please point out which cells are considered to be spermatogonia and spermatocytes in the mutant.

We have put new representative pictures where we can more clearly observe the spermatogonia and spermatocytes. Also, we have indicated spermatogonia (SG), spermatocyte (SC), and spermatid (SP) by arrowheads.

14. Line 247: It is likely that MEIOB accumulates in foci in the mutant but it is nevertheless not shown, so MEIOB should be removed from this line.

We have removed MEIOB according to your suggestion.

line 245-247

“These data suggest that the incompletely repaired recombination intermediates, which are associated with SPATA22, accumulated in Meilb2^{-/-} spermatocytes.”

15. Line 271: Variation can exist between animals at this stage. How many embryos were analyzed?

We have clearly mentioned the number of embryos and mice used for this quantification in the figure legend of Supplementary Fig. 6.

Supplementary Fig. 6.

*“The mean values of four independent experiments for (+/+) (from four embryos from four individual mice) and two independent experiments for (-/-) (from two embryos from two individual mice) are shown. Error bars show the SD. All analyses were with two-tailed t-tests. N.S., not significant. *p < 0.05.”*

16. Supp Fig 7b: corrupted figure: Data not visible in the graph.

This was not corrupted figure. All Metaphase I oocytes showed 20 bivalent chromosomes both in WT and *Meilb2* KO, so that all dot are aligned horizontally. To make it more visible, we have stretched the graph in a horizontal direction.

17. Line 305: This sentence contradicts the next (or at least is confusing); from the cited paper it is clear that *Brca2* nullizygosity leads to meiotic arrest in males and that *Meilb2* mutant We are thinking that our sentences (as shown below) are appropriate.

Line 306-308

Brca2 KO mice are embryonic lethal, and thus there has been **no direct** assessment of this gene's function in meiosis. However, *Brca2*-null mice expressing human *BRCA2* rescued the embryonic lethality and showed sterility with reduced localization of recombinases similar to our *Meilb2*^{-/-} mice.

The *Brca2* nullizygosity mice (*Brca2* KO mice) is embryonic lethal and there has been no direct assessment of its meiotic phenotype in this KO mice. However, *Brca2*-null mice expressing human *BRCA2* rescued the embryonic lethality and showed sterility. The “*Brca2* nullizygosity mice (*Brca2* KO mice)” is not equal to the “*Brca2*-null mice expressing human *BRCA2*”, so that we are thinking that our sentences are correct.

18. Line 413: The authors suggest that the complete loss of RAD51/DMC1 foci in males is due to a stronger checkpoint function in females. I am not aware of checkpoints regulating focus formation. This needs to be clarified

We are sorry for the confusing sentences but we were not trying to mean that “the complete loss of RAD51/DMC1 foci in males is due to a stronger checkpoint function in females”.

We are trying to mean that “the sexual dimorphic phenotype reported in several recombination mutant mice was thought to be due to the weaker checkpoint-like mechanism in female.”

To avoid confusion, we have deleted the sentence mentioning this putative checkpoint activity.

Before

*“The sexual dimorphic phenotype has been reported in a number of meiotic recombination mutant mice, where female meiocytes always reach more advanced stages compared to those in males. The reason for this sexual dimorphism is still a matter of debate, but it is speculated that oocytes have a weaker checkpoint-like mechanism that eliminates dysfunctional meiocytes during prophase I progression. This is also the case for our *Meilb2* KO mice.”*

After (line 411-413)

*“The sexual dimorphic phenotype has been reported in a number of meiotic recombination mutant mice, where female meiocytes always reach more advanced stages compared to those in males. This is also the case for our *Meilb2* KO mice.”*

19. Line 430: The authors refer to the CCLE to say that *Meilb2* upregulation is observed in cancers, but not analysis is provided for this point.

The data showing the upregulation of *Meilb2* in cancer cells are available in Cancer Cell Line Encyclopedia (CCLE), a publicly available database. This is not our experimental data, but from a publicly available database, so that we put it in discussion section. We are thinking that having a discussion referring to the publicly available database is acceptable in the discussion section.

20. Fig. 6f. In the model, RPA recruits MEILB2. This is confusing since MEILB2 does not interact with RPA, and SPATA22, but not RPA, accumulate in *Meilb2*^{-/-}.

We agree with your point.

1, We showed that MEILB2 interacts with SPATA22 *in vivo* (Supplementary Fig. 5d).

2, It is already established that SPATA22 forms complex with MEIOB as well as RPA complex in the preceding studies.

Accordingly, we have revised the model as below, showing that RPA-SPATA22-MEIOB complex is likely the scaffold of MEILB2.

REVIEWERS' COMMENTS:

Reviewer #3 (Remarks to the Author):

The authors have addressed most of the comments. Here are some final points that apparently were not clear in the previous review.

2. The authors are correct that MEI is frequently used to name new meiotic genes. However, besides being confusing with MEIOB, the other concern is that the gene is Hsf2bp in the database. It will be difficult to keep track of papers that use one name for meiotic studies and another name in cancer and ES cell studies. At the very least, Hsf2bp should be introduced into the abstract when Meilb2 is first mentioned.

7. It is unusual to graph individual testis weights. The authors should combine the two testes from one mouse so there will be 3 data points rather than 6.

9. The point is that the criteria for staging should be included in the manuscript. Prophase I substages are in Fig. 1d but the criteria is not in the legend or elsewhere, as far as I could see.

19. Readers should not have to go to CCLE to figure out what data the authors are specifically referring to. It is acceptable to publish data derived from publicly available databases with attribution.

Please find below our point-by-point responses to the requests.

Reviewer comment #2. The authors are correct that MEI is frequently used to name new meiotic genes. However, besides being confusing with MEIOB, the other concern is that the gene is Hsf2bp in the database. It will be difficult to keep track of papers that use one name for meiotic studies and another name in cancer and ES cell studies. At the very least, Hsf2bp should be introduced into the abstract when Meilb2 is first mentioned.

We have added *Hsf2bp* in the abstract, where Meilb2 is first mentioned.

Reviewer comment #7. It is unusual to graph individual testis weights. The authors should combine the two testes from one mouse so there will be 3 data points rather than 6.

We have combined testes weight in Fig. 3c and Supplementary Fig. 4a.

Reviewer comment #9. The point is that the criteria for staging should be included in the manuscript. Prophase I substages are in Fig. 1d but the criteria is not in the legend or elsewhere, as far as I could see.

We have added the stage definition in figure legend of Fig. 1d.

Reviewer comment #19. Readers should not have to go to CCLE to figure out what data the authors are specifically referring to. It is acceptable to publish data derived from publicly available databases with attribution.

We have added CCLE data in Supplementary Fig.9.

Hiroki Shibuya

Department of Chemistry and Molecular Biology, University of Gothenburg, SE-40530

Gothenburg, Sweden.

hiroki.shibuya@gu.se